# Scalable Long Range Propagation on Continuous-Time Dynamic Graphs

## Abstract

Recent research on Deep Graph Networks (DGNs) has broadened the domain of learning on graphs to real-world systems of interconnected entities that evolve over time. This paper addresses prediction problems on graphs defined by a stream of events, possibly irregularly sampled over time, generally referred to as Continuous-Time Dynamic Graphs (C-TDGs). While many predictive problems on graphs may require capturing interactions between nodes at different distances, existing DGNs for C-TDGs are not designed to propagate and preserve long-range information – resulting in suboptimal performance. In this work, we present *Continuous-Time Graph Anti-Symmetric Network* (CTAN), a DGN for C-TDGs designed within the ordinary differential equations framework that enables efficient propagation of long-range dependencies. We show that our method robustly performs stable and non-dissipative information propagation over dynamically evolving graphs, where the number of ODE discretization steps allows scaling the propagation range. We empirically validate the proposed approach on several real and synthetic graph benchmarks, showing that CTAN leads to improved performance while enabling the propagation of long-range information.

## 1 Introduction

Graphs are a highly expressive abstraction for modelling entities and their relations, e.g., molecular structures, recommender systems, or traffic networks. Deep Graph Networks (DGNs) (Bacciu et al., 2020; Wu et al., 2021) have lately emerged as a family of deep learning models that can effectively process and learn such structured information. While most of the proposed DGNs have been designed for static graphs, many real-world scenarios are inherently *dynamic* in nature. Examples include the continual activities and interactions between members of social as well as communication networks, recurrent purchases by users on e-commerce platforms, or evolving interactions of processes with files in an operating system. The community has therefore begun to investigate models that can process the temporal dimension of a dynamic graph (Kazemi et al., 2020), with more recent interest into those graphs defined through a stream of events (possibly irregularly sampled over time), known as Continuous-Time Dynamic Graphs (C-TDGs). However, given that the newly proposed methods are designed upon static DGNs, they inherit all their weaknesses. Specifically, they suffer from the *over-squashing* phenomenon (Alon & Yahav, 2021), which prevents the final DGN to learn and propagate long range information (Gravina et al., 2023). With growing evidence from the static case (Dwivedi et al., 2022) that long-range dependencies are necessary for effective learning, the ability to learn properties beyond local neighborhoods remains an open challenge in the C-TDG domain.

In this paper, we propose the Continuous-Time Graph Anti-Symmetric Network (CTAN), a framework for learning of C-TDGs with scalable long range propagation of information, thanks to properties inherited from stable and non-dissipative ordinary differential equations (ODEs). We establish theoretical conditions for achieving stability and non-dissipation in the CTAN ODE by employing anti-symmetric weight matrices, which is the key factor for modeling long-range spatio-temporal interactions. The CTAN layer is derived from the forward Euler discretization of the designed differential equation. The formulation of CTAN allows scaling the radius of propagation of information depending on the number of discretization steps, i.e., the number of layers in the final architecture. Remarkably, even with a limited number of layers, the non-dissipative behavior enables the transmission of information for a past event as new events occur, since node states are used to effi-

ciently retain and propagate historical information. Consequently, this mechanism permits scaling the single event propagation to cover a larger portion of the C-TDG. The general formulation of the node update state function allows extending DGNs for static graphs to the domain of C-TDGs, thus reinterpreting current state-of-the-art static DGNs as a discretized representation of non-dissipative ODEs tailored for C-TDGs. To the best of our knowledge, CTAN is the first framework to address the problem of long-range propagation in C-TDGs and the first to bridge the gap between ODEs and C-TDGs.

The key contributions of this work can be summarized as follows: (i) We introduce the problem of long-range propagation (i.e., non-dissipativeness) within C-TDGs; (ii) We introduce CTAN, a new deep graph network for learning C-TDGs based on ODEs, which enables stable and non-dissipative propagation to preserve long term dependencies in the information flow, and it does so in a theoretically founded way; (iii) we present novel benchmark datasets specifically designed to assess the ability of DGNs to propagate information over long spatio-temporal distances within C-TDGs; (iv) we conduct extensive experiments to demonstrate the benefits of our method, showing that CTAN not only outperforms state-of-the-art DGNs on synthetic long-range tasks but also outperforms them on several real-world benchmark datasets.

## 2 PRELIMINARIES

We consider a *dynamic graph* as the tuple $\mathcal{G}(t) = (\mathcal{V}(t), \mathcal{E}(t), \mathbf{X}(t), \mathbf{E}(t))$, defined for any time $t \geq 0$, which models a dynamical system of interacting entities (also known as *nodes*) where interactions (or *edges*) evolve over time, i.e., they are dynamic in nature. Here, $\mathcal{V}(t)$ is the set of nodes that are present in the graph at time $t$, and $\mathcal{E}(t) \subseteq \{\{u, v, t^-\} \mid u, v \in \mathcal{V}(t), t^- < t\}$ defines the edges between them. Matrices $\mathbf{X}(t) \in \mathbb{R}^{|\mathcal{V}(t)| \times d_n}$ and $\mathbf{E}(t) \in \mathbb{R}^{|\mathcal{E}(t)| \times d_e}$ contain node and edge features, respectively. The $u$-th row of $\mathbf{X}(t)$ is denoted as $\mathbf{x}_u$ and represents the features of the single node $u$. Similarly, we indicate edge feature vectors as $\mathbf{e}_{uvt}$. Each node $u$ is also associated to state $\mathbf{h}_u(t) \in \mathbf{H}(t) \in \mathbb{R}^{|\mathcal{V}(t)| \times d}$, which encodes node evolution over time $t$.

In our setting, the dynamic graph is observed as a stream of events, also known as observations, that can appear irregularly over time. Therefore, the system of interacting entities is not fully observed over time, and it is known as C-TDG. In this scenario, the dynamic graph can be rewritten as $\mathcal{G} = \{o_t \mid t \in [t_0, t_n]\}$, where each event $o_t = (t, EventType, u, v, \mathbf{x}_u, \mathbf{x}_v, \mathbf{e}_{uvt})$ is a tuple containing information regarding the timestamp, the event type, the involved nodes, and their states. The event types can be grouped into three main classes, which are node-wise events (i.e., a node is updated or created), interaction events (i.e., an edge is created), and deletion events (i.e., a node/edge is deleted). In the following, we will refer to $\mathcal{V}_\oplus$ as the event "node creation", and to $\mathcal{E}_\oplus$ as "edge addition". We present in Figure 1 a visual exemplification of a C-TDG.

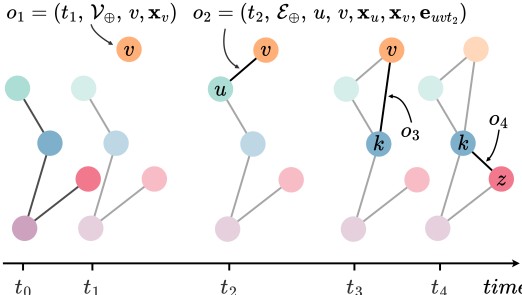

Figure 1: The evolution of a Continuous-Time Dynamic Graph through the stream of events up to timestamp $t_4$. The event type $\mathcal{V}_\oplus$ corresponds to "node creation", and $\mathcal{E}_\oplus$ means "edge addition". At each timestamp, the faded portion of the graph corresponds to historical information.

## 3 CONTINUOUS-TIME GRAPH ANTI-SYMMETRIC NETWORK

Learning the dynamics of a C-TDG can be cast as the problem of learning information propagation following newly observed events in the system. This entails learning a diffusion function that updates the state of node $u$ as

$$\mathbf{h}_u(t) = F\left(t, \mathbf{x}_u, \mathbf{h}_u(t), \{\mathbf{h}_v(t)\}, \{\mathbf{e}_{uvt^-}\}\right), \quad \text{with } (v, t^-) \in \mathcal{N}_u^t \tag{1}$$

where $\mathcal{N}_u^t = \{(v, t^-) \mid \{u, v, t^-\} \in \mathcal{E}(t)\}$ is the temporal neighborhood of a node $u$ at time $t$, which consists of all the historical neighbors of $u$ prior to current time $t$. In the following, we omit the time

subscript from the edge feature vector to enhance readability, since it refers to a time in the past in which the edge appeared.

In recent literature, Eq. 1 is modeled through a dynamical system described by a learnable ordinary differential equation (ODE) (Poli et al., 2019; Chamberlain et al., 2021; Eliasof et al., 2021; Rusch et al., 2022; Gravina et al., 2023). Differently from discrete models, neural-ODE-based approaches learn more effective latent dynamics and have shown the ability to learn complex temporal patterns from irregularly sampled timestamps (Chen et al., 2018; Rubanova et al., 2019; Kidger et al., 2020), making them more suitable to address C-TDG problems.

In this paper, we leverage non-dissipative ODEs (Haber & Ruthotto, 2017; Chang et al., 2019; Gravina et al., 2023) for the processing of C-TDGs. Thus, we propose a framework as a solution to a *stable* and *non-dissipative* ODE over a streamed graph. The main goal of our work is therefore achieving preservation of long-range information between nodes over a stream of events. We do so by first showing how a generic ODE can learn the hidden dynamics of a C-TDG and then by deriving the condition under which the ODE is constrained to the desired behavior.

First, we define a Cauchy problem in terms of the following node-wise ODE defined in time $t \in [0, T]$

$$\frac{\partial \mathbf{h}_u(t)}{\partial t} = f_\theta \left( t, \mathbf{x}_u, \mathbf{h}_u(t), \{\mathbf{h}_v(t)\}_{v \in \mathcal{N}_u^t}, \{\mathbf{e}_{uv}\}_{v \in \mathcal{N}_u^t} \right) \tag{2}$$

and subject to an initial condition $\mathbf{h}_u(0) \in \mathbb{R}^d$. The term $f_\theta$ is a function parametrized by the weights $\theta$ that describes the dynamics of node state. We observe that this framework can naturally deal with events that arrive at an arbitrary time. Indeed, the original Cauchy problem in Eq. 2 can be divided into multiple sub-problems, one per each event in the C-TDG. The $i$-th sub-problem, defined in the interval $t \in [t_s, t_e]$, is responsible for propagating only the information encoded by the $i$-th event. Overall, when a new event $o_i$ happens, the ODE in Eq. 2 computes new nodes' representations, $\mathbf{h}_u^i(t_e)$, starting from the initial configurations, $\mathbf{h}_u^i(t_s)$. In other words, $f_\theta$ evolves the state of each node given its initial condition. The top-right of Figure 2 visually summarizes this concept, showing the nodes' evolution given the propagation of an incoming event. We observe that the knowledge of past events is preserved and propagated in the system thanks to an initial condition that includes not only the current nodes' input state but also the nodes' representations computed in the previous sub-problem, i.e., $\mathbf{h}_u^i(t_s) = \psi(\mathbf{h}_u^{i-1}(t_e), \mathbf{x}_u(i))$. We notice that the terminal time $t_e$ is responsible for determining the extent of information propagation across the graph, since it limits the propagation to a constrained distance from the source. Consequently, smaller values of $t_e$ allow only for localized event propagation, whereas larger values enable the dissemination of information to a broader set of nodes.

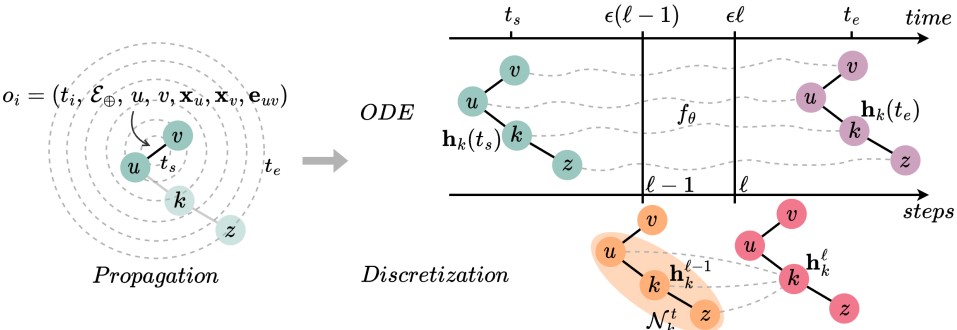

Figure 2: A high-level overview of the proposed framework illustrated for the $i$-th Cauchy sub-problem. On the left, we depict the propagation of the information of event $o_i$ through the graph. The faded portion of the graph corresponds to historical information, while the rest is the incoming event. On the right, we illustrate the evolution of nodes' states given the propagation of the incoming event. Specifically, the top right shows the evolution as an ODE, $f_\theta$, that computes the node representation for a node $k$, $\mathbf{h}_k(t)$. Such computation is subject to an initial condition $\mathbf{h}_k(t_s) = \psi(\mathbf{h}_k^{i-1}(t_e), \mathbf{x}_k(i))$ that includes the node representations computed in the previous sub-problem and the current node input state. In the bottom right, the discretized solution of the ODE is computed as iterative steps of the method over a discrete set of points in the time interval $[t_s, t_e]$.

We now proceed to derive the condition under which the ODE is constrained to the stable and non-dissipative behavior, allowing for the propagation of long-range dependencies in the information flow. We start by defining the concepts of non-dissipativeness[1] as follow

**Definition 1** (Non-dissipativeness over a C-TDG). *Let $u, v \in \mathcal{V}(t)$ be two nodes of the C-TDG at some time $t$, connected by a path of length $L$. If an event $o_i$ occurs at node $u$, then $o_i$'s information is propagated from $u$ to $v$, $\forall L \geq 0$.*

We start by instantiating Eq. 2 as

$$\frac{\partial \mathbf{h}_u(t)}{\partial t} = \sigma \left( \mathbf{W}_t \mathbf{h}_u(t) + \Phi \left( \{\mathbf{h}_v(t), \mathbf{e}_{uv}, t_v^-, t\}_{v \in \mathcal{N}_u^t} \right) + \mathbf{b}_t \right) \tag{3}$$

where $\sigma$ is a monotonically non-decreasing activation function; $\Phi$ is the aggregation function that computes the representation of the neighborhood of the node $u$ considering nodes' states and edges' features; $t_v^-$ is the time point of the previous event for node $v$; and $\mathbf{W}_t \in \mathbb{R}^{d \times d}$ and $\mathbf{b}_t \in \mathbb{R}^d$ are the parameters of the system. We notice that including $t_v^-$ in $\Phi$ has the role of encoding the time elapsed since the previous event involving node $v$. This represents a force that may smoothly update the node's current state during the time interval to prevent the staleness problem (Kazemi et al., 2020).

As discussed in Haber & Ruthotto (2017) and Gravina et al. (2023), a non-dissipative propagation is directly linked to the sensitivity of the solution of the ODE to its initial condition, thus to the stability of the system. Such sensitivity is controlled by the Jacobian's eigenvalues of Eq. 3. Given $\lambda_i(\mathbf{J}(t))$ the $i$-th eigenvalue of the Jacobian, when $\max_{i=1,\dots,d} Re(\lambda_i(\mathbf{J}(t))) > 0$, a small perturbation of the initial condition would cause an exponentially exploding difference in the nodes representations, and the system would be unstable. On the contrary, for $\max_{i=1,\dots,d} Re(\lambda_i(\mathbf{J}(t))) \ll 0$, the difference in the nodes' representations would vanish exponentially fast over time, thereby making the nodes' representation insensitive to differences in the initial condition. This results in a lossy system in which only local neighborhood information is preserved, while long-range dependencies among nodes are forgotten. Consequently, the resulting system is dissipative. Lastly, if $Re(\lambda_i(\mathbf{J}(t))) = 0$ for $i = 1, \dots, d$ the initial condition is effectively propagated into the final node representation, making the system both stable and non-dissipative[2]. The next proposition ensures that the employment of anti-symmetric weight matrices constrains the Jacobian's eigenvalues to be imaginary, allowing a stable and non-dissipative behavior of the ODE in Eq. 3.

**Proposition 1.** *Provided that the weight matrix $\mathbf{W_t}$ is anti-symmetric[3] and the aggregation function $\Phi$ does not depend on $\mathbf{h}_u(t)$, the Jacobian matrix resulting from the ODE in Eq. 3 has purely imaginary eigenvalues, i.e.,*

$$Re(\lambda_i(\mathbf{J}(t))) = 0, \forall i = 1, \dots, d.$$

See proof in Appendix A. A more in-depth analysis about non-dissipativeness is in Appendix B.

Now that we have defined the conditions under which the ODE in Eq. 3 is stable and non-dissipative, i.e., it can propagate long-range dependencies between nodes in the C-TDG, we observe that computing the analytical solution of an ODE is usually infeasible. It is common practice to rely on a discretization method to compute an approximate solution by multiple applications of the method over a discrete set of points in the time interval $[t_s, t_e]$. This process is visually summarized at the bottom of Figure 2. For simplicity, we employ the *forward Euler's method* to discretize Eq. 3 for the $i$-th Cauchy sub-problem, yielding the following node state update equation for the node $u$ at step $\ell$:

$$\mathbf{h}_u^\ell = \mathbf{h}_u^{\ell-1} + \epsilon \sigma \left( (\mathbf{W}_\ell - \mathbf{W}_\ell^\top - \gamma \mathbf{I}) \mathbf{h}_u^{\ell-1} + \Phi \left( \{\mathbf{h}_v(t), \mathbf{e}_{uv}, t_v^-, t\}_{v \in \mathcal{N}_u^t} \right) + \mathbf{b}_\ell \right), \tag{4}$$

with $\epsilon > 0$ being the discretization step size. We notice that the anti-symmetric weight matrix $(\mathbf{W}_\ell - \mathbf{W}_\ell^\top)$ is subtracted by the term $\gamma \mathbf{I}$ to preserve the stability of the forward Euler's method, see Appendix C for a more in-depth analysis. We refer to $\mathbf{I}$ as the identity matrix and $\gamma$ to a hyper-parameter that regulates the stability of the discretized diffusion. We note that the resulting neural architecture contains as many layers as the discretization steps, i.e., $L = t_e / \epsilon$.

---

[1] The interested reader is referred to (Glendinning, 1994; Ascher et al., 1995) for an in-depth analysis of dissipative dynamical systems.

[2] This result holds also when the eigenvalues of the resulting Jacobian are still bounded in a small neighborhood around the imaginary axis (Gravina et al., 2023).

[3] A matrix $\mathbf{M} \in \mathbb{R}^{d \times d}$ is anti-symmetric (i.e., skew-symmetric) if $\mathbf{M}^\top = -\mathbf{M}$.

As previously discussed for the ODE, the number of iterations in the discretization (i.e., the terminal time $t_e$) plays a crucial role in the propagation. Specifically, few iterations result in a localized event propagation. Consequently, the non-dissipative event propagation does not reach each node in the graph, causing a *truncated* non-dissipative propagation. This method allows scaling the radius of propagation of information depending on the number of discretization steps, thus allowing for a scalable long-range propagation in C-TDGs. Crucially, we notice that, even with few discretization steps, it is still possible to propagate information from a node $u$ to $z$ (if a path of length $P$ connects $u$ and $z$). As an example, let's consider the situation depicted in the left segment of Figure 2, where nodes $u$ and $v$ establish a connection at some time $t$, and our objective is to transmit this information to node $z$. In this scenario, we assume $L = 1$, thus the propagation is truncated before $z$. Upon the arrival of the event at time $t$, this is initially relayed (due to the constraint of $L = 1$) to node $k$, which then captures and retains this information. If a future event at time $t + \tau$ involving node $k$ occurs, its state is propagated, ultimately reaching node $z$. Consequently, the information originating from node $u$ successfully traverses the structure to reach node $z$. More formally, if it exists a sequence of (at least $P/L$) successive events, such that each future $i$-th event is propagated to an intermediate node at distance $iP/L$ from $u$, then $u$ is able to directly share its information with $z$. Therefore, even with a limited number of discretization steps, the non-dissipative behavior enables scaling the single event propagation to cover a larger portion of the C-TDG. We also notice that if the number of iterations is at least equal to the longest shortest path in the C-TDG, then each event is always propagated throughout the whole graph.

We name the overall framework defined above Continuous-Time Graph Anti-Symmetric Network (CTAN). Note that $\Phi$ in Eq. 3 and 4 can be any function that aggregates nodes and edges states. Then, CTAN can leverage the aggregation function that is more adequate for the specific task. As an exemplification of this, in Section 5 we leverage the aggregation scheme based on the one proposed by Shi et al. (2021):

$$\Phi\left(\{\mathbf{h}_v(t), \mathbf{e}_{uv}, t_v^-, t\}_{v \in \mathcal{N}_u^t}\right) = \sum_{v \in \mathcal{N}_u^t \cup \{u\}} \alpha_{uv}\left(\mathbf{V}_n \mathbf{h}_v^{\ell-1} + \mathbf{V}_e \hat{\mathbf{e}}_{uv}\right) \tag{5}$$

where $\hat{\mathbf{e}}_{uv} = \mathbf{e}_{uv} \| \left(\mathbf{V}(t - t_v^-)\right)$ is the new edge representation computed as the concatenation between the original edge attributes and a learned embedding of the elapsed time from the previous neighbor interaction, and $\alpha_{uv} = \text{softmax}\left(\frac{\mathbf{q}^\top \mathbf{K}}{\sqrt{d}}\right)$ is the attention coefficient with $d$ the hidden size of each head, $\mathbf{q} = \mathbf{V}_q \mathbf{h}_u^{\ell-1}$, and $\mathbf{K} = \mathbf{V}_k \mathbf{h}_v^{\ell-1} + \mathbf{V}_e \hat{\mathbf{e}}_{uv}$. We use $\left(\mathbf{V}(t - t_v^-)\right)$ as a force that smoothly updates the node's current state to prevent the staleness problem.

Despite CTAN being designed from the general perspective of layer-dependent weights, it can easily used as a recursive method with weight sharing between layers. In Section 5 we use weight sharing to prove the efficacy of our method.

## 4 RELATED WORK

**Deep Graph Network for C-TDGs.** Nowadays, most of the DGNs tailored for learning C-TDGs can be generalized within the Temporal Graph Network (TGN) framework (Rossi et al., 2020). This architecture comprises three main modules: a message module, which is responsible for computing a message that encodes the incoming event; a memory module, which stores the node's history; and a graph propagation module, which aggregates information from the local neighborhood to produce the final node representation. Usually, the memory module is implemented as a Recurrent Neural Network (RNN) and the graph propagation module as a DGN for the processing of static graphs. Many state-of-the-art architectures (Kumar et al., 2019; Trivedi et al., 2019; Xu et al., 2020; Ma et al., 2020; Souza et al., 2022) fit this framework, with later methods outperforming earlier ones thanks to advances in the local message passing part or even in the encoding of positional features. While recent methods often provide improved results, none of them explicitly models long-range dependencies between nodes or events in the C-TDG. As increasingly evidenced both in sequence-model architectures (Chang et al., 2019), and in the static graph case (Dwivedi et al., 2022), propagating information across various time steps is extremely beneficial for learning; also as evidenced by the long-range experiments in Section 5.

CTAN, instead, provably enables effective long-range propagation by design. Note that our approach does not require the co-existence of memory *and* graph propagation module, as in the TGN

framework. CTAN stores all necessary information within the node embeddings themselves as computed by the graph convolution, while achieving non-dissipative propagation by design. This makes CTAN a more lightweight and faster architecture. Lastly, as TGN allows for different graph propagation modules, the general formulation of the aggregation function $\Phi$ in Eq. 4 allows extending state-of-the-art DGNs for static graphs to the domain of C-TDGs through the lens of non-dissipative and stable ODEs.

**Continuous Dynamic Models.** Neural Differential Equations have emerged as a class of neural networks suitable for learning continuous dynamics of systems. Chen et al. (2018) and Chang et al. (2019) parameterize the continuous dynamic of RNNs through an ordinary differential equation. Similarly, Gallicchio (2022) draws a connection with Reservoir Computing. Despite the similarity with RNNs, such architectures have shown the ability to naturally incorporate data that arrive at arbitrary times (Chen et al., 2018; Rubanova et al., 2019). Inspired by the NeuralODE approach, GDE (Poli et al., 2019) links DGNs for static graphs with ODEs. In this scenario, the inter-layer dynamic of DGN's node representation is designed as a continuous information processing system defined by an ODE, which, starting from the input configuration of the nodes' states, computes the final nodes' representations. In the static graph domain, ODE-based architectures have been proposed with different aims, such as reducing the computational complexity of message passing (Wu et al., 2019; Wang et al., 2021), or mitigating the over-smoothing phenomena (Eliasof et al., 2021; Rusch et al., 2022).

To the best of our knowledge, we are the first to propose an ODE-based architecture suitable for C-TDGs that can effectively propagate long-range information between nodes.

## 5 EXPERIMENTS

We evaluate CTAN on two novel tasks to show the efficacy of propagating long-range information between nodes in Section 5.1.1 and Section 5.1.2. Moreover, we assess the performance of the proposed CTAN approach on classical benchmarks for C-TDGs in Section 5.2 and Appendix G. In Appendix H we conduct an investigation on the scalability property of CTAN. In Appendix D, we present comprehensive descriptions and statistics of the employed datasets, accompanied by an analysis of the tasks' inductiveness. We release openly the code implementing our methodology and reproducing our empirical analysis[4].

**Shared Experimental Settings.** In the following experiments, we consider weight sharing of CTAN parameters across the neural layers. We compare CTAN against four popular dynamic graph network methods: DyRep (Trivedi et al., 2019), JODIE (Kumar et al., 2019), TGAT (Xu et al., 2020), and TGN (Rossi et al., 2020). These related methods are used across all tasks considered. To ensure fair comparison and efficient implementation, we re-implement these methods in our framework, based on the original works' repositories. With the same purpose, we reused the graph convolution operators in the original literature, considering for all methods the aggregation function defined in Eq. 5. We designed each model as a combination of two components: (i) the DGN (i.e., CTAN or a baseline) which is responsible to compute the nodes' representations; (ii) the readout that maps the output of the DGN into the output space. The readout is a 2-layer MLP, used in all models with the same architecture. We performed rigorous hyper-parameter tuning via grid search, considering a fixed parameter budget based on the number of graph convolutional layers (GCLs). Specifically, for the maximum number of GCL in the grid, we select the embedding dimension so that the total number of parameters matches the budget; such embedding dimension is used across every other configuration. We report more detailed information on each task in their respective subsections in the following. Detailed information about hyper-parameter grids and training of models are reported in Appendix E. We note that, in the following subsections, while we not directly investigate the optimal terminal time $t_e$ within the hyper-parameter space, we implicitly address this aspect through the choice of the step size $\epsilon$ and the maximum number of layers $L$, as they jointly determine the terminal time, i.e., $t_e = \epsilon L$.

---

[4]Upon acceptance

## 5.1 LONG RANGE TASKS

Given the lack of available datasets containing long-range interactions in the C-TDG setting, we introduce two *temporal* tasks which contain long-range interaction (LRI). The first is a classification task on path graphs (Bondy & Murty, 1976) and the second an extension to the temporal domain of the classification task `PascalVOC-SP` introduced in the Long Range Graph Benchmark (Dwivedi et al., 2022).

### 5.1.1 SEQUENCE CLASSIFICATION ON TEMPORAL PATH GRAPH

**Setup.** We consider a classification task on a temporal interpretation of a path graph (Bondy & Murty, 1976). In our interpretation, the nodes of the path graph appear sequentially over time from first to last, i.e., each event in the C-TDG connects each node to the previous one in the path graph (see Appendix D for a reference to our dataset). The objective of the task is to predict the feature associated to the source node in the first event after having traversed the entire temporal path graph, i.e., after reaching the last event in the stream. This task corresponds to predicting the information attached to the first node having appeared in the network, after propagating it (incrementally, in time) through the entire C-TDG, making the task similar to a classical sequence classification task. To be consistent with the setup used in the C-TDG domain, events are forwarded one at time to update neighboring nodes representations. After the model processes the last event in the graph, the output prediction for the whole graph is computed by a readout that takes as input *only* the updated embedding of the destination node of the last event in the C-TDG.

We set the feature of the first source node to be either 1 or -1, and we use random features for intermediate nodes and edges to make the task more complex. In this task it is fundamental to propagate the initial information through the entire graph to make accurate predictions, thus capturing long-range interactions.

We considered graphs of different sizes (from paths of length 3 to 20) to test how long information is propagated, i.e., the longer the graph, the more the model needs to be capable of propagating information over time to complete the task. During training, we optimize the binary cross-entropy loss over two classes corresponding to the two possible signals (1 or-1) placed on the initial node. Each experimental run is repeated 10 times with different random seeds to ensure multiple weight initializations. The best performing configuration is chosen based on the validation loss. We report in Appendix E more training details and the grid of hyper-parameters employed for this experiment. We observe that the grid is computed considering a budget of ∼20k trainable parameters per model.

**Results.** The test accuracy on the path graph task is provided in Figure 3 (and Table 5 in tabular form). We notice that CTAN exhibits exceptional performance in comparison to reference state-of-the-art methods. More interestingly, our method achieves consistent results across different path graph lengths.

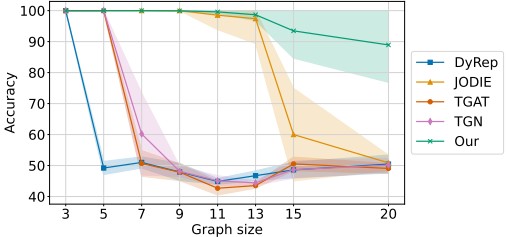

Figure 3: Mean test set accuracy with respect to the temporal path graph size. Results are averaged over 10 random weight initializations.

This result highlights the capability of our method to propagate information seen on the first node throughout long paths, while baseline models struggle in solving such a task because the information is lost through the timesteps, i.e., in practice, informative gradients vanish over time. All literature methods experience sharp drops in performance, bringing them closer to the performance level of a random guesser as path size increases. We believe that the difficulty of baselines in information propagation lies in their inherent limitations. TGAT, for instance, lacks a dedicated memory mechanism and instead computes projections of input features, restricting its information propagation range to the number of layers present (we test a maximum of 5 layers, which matches the longest path graph where TGAT can solve the task). Differently, DyRep relies on a single layer of graph convolution, which makes it susceptible to over-squashing. The only competitive model is JODIE, which is capable of effectively propagating information up to a graph size equal to 13. Nonetheless, as known from related works (Bengio et al., 1994; Chang et al., 2019), RNN-based architectures

like JODIE inherently struggle to maintain long-term dependencies, affecting their ability to propagate information in C-TDGs. These results demonstrate that CTAN better captures and models long range information in C-TDGs.

### 5.1.2 CLASSIFICATION ON TEMPORAL PASCAL-VOC

**Setup.** As a second LRI task, we consider edge classification on a temporal interpretation of the `PascalVOC-SP` dataset, which has been previously employed by Dwivedi et al. (2022) as a benchmark to show the efficacy of capturing LRI in static graphs. Here, we adapt the task to the C-TDG domain. Specifically, we forward edges one at a time and predict the class of the destination node. We generate temporal graphs starting from the dataset of `rag-boundary` graphs extracted from Pascal VOC 2011 provided in Dwivedi et al. (2022) (more details are provided in Appendix D). We consider two degrees of SLIC superpixels compactness, i.e., 10 and 30. We observe that larger compactness means more patches, with less information included in each patch and more to be propagated.

During training, we optimize the F1-score as in Dwivedi et al. (2022). To benchmark the ability of models to propagate information through the graph, we test model performance for an increasing number of GCLs. Fewer graph convolutions require models to store and transmit relevant information along node embeddings rather than relying on effectively aggregating information from increasingly larger neighborhoods. Each experimental run is repeated 5 times with different random seeds to consider multiple weight initializations. Best configurations are chosen based on the validation loss (cross-entropy). Appendix E provides further training and model selection details. We observe that the grid is computed considering a budget of trainable parameters per model equal to $\sim$40k.

**Results.** Table 1 reports the average F1-score on the Temporal `PascalVOC-SP` test. Note that DyRep and JODIE, in their original definition, do not support a variable number of GCLs, hence the results of such models are presented under "1 GCL" in the tables for simplicity.

Table 1: Results of the classification on the Temporal `PascalVOC` task, for increasing number of GCLs. The performance metric is the mean test set F1-score, averaged over 5 different random weights initializations for each model configuration. Models have a maximum budget of learnable parameters equal to $\sim$40k.

| no. GCLs | Temporal Pascal VOC (sc=10) | | | Temporal Pascal VOC (sc=30) | | |
|---|---|---|---|---|---|---|
| | 1 | 3 | 5 | 1 | 3 | 5 |
| DyRep | $5.29_{\pm 0.47}$ | - | - | $5.23_{\pm 0.11}$ | - | - |
| JODIE | $6.33_{\pm 0.41}$ | - | - | $5.76_{\pm 0.35}$ | - | - |
| TGAT | $5.39_{\pm 0.19}$ | $6.53_{\pm 0.58}$ | $8.23_{\pm 0.73}$ | $6.04_{\pm 0.26}$ | $8.79_{\pm 0.29}$ | $\mathbf{10.38}_{\pm 0.7}$ |
| TGN | $6.04_{\pm 0.27}$ | $6.55_{\pm 0.46}$ | $7.51_{\pm 0.8}$ | $5.59_{\pm 0.24}$ | $7.26_{\pm 0.82}$ | $7.9_{\pm 1.31}$ |
| Our | $\mathbf{7.89}_{\pm 0.33}$ | $\mathbf{8.53}_{\pm 1.06}$ | $\mathbf{8.88}_{\pm 0.98}$ | $\mathbf{9.98}_{\pm 0.33}$ | $\mathbf{10.16}_{\pm 0.52}$ | $10.36_{\pm 0.47}$ |

CTAN largely outperforms reference methods for smaller GCL numbers, even with sizeable improvements. We observe that for SLIC compactness equal to 30, CTAN achieves a 65% and 16% improvement against the 2$^{nd}$ best performing model (i.e., TGAT), for one and three GCLs, respectively. Interestingly, TGAT slightly outperforms CTAN in one setup when considering five GCLs. This is in line with the excellent results of computationally expensive Transformers-based models in the static case (Dwivedi et al., 2022), corroborating the advantages of self-attention blocks in modeling long-range dependencies between far away nodes. This result also suggests that the majority of the relevant information necessary to solve the Temporal `Pascal VOC` task may lie within neighborhoods five hops away. Nevertheless, the results indicate how CTAN is capable of propagating relevant information across the time-steps to achieve accurate predictions, even when the model is only allowed to extract information from limited, very local neighborhoods.

### 5.2 BENCHMARK TASKS

**Setup.** For the C-TDG benchmarks we consider four well-known datasets proposed by Kumar et al. (2019), which are Wikipedia, Reddit, LastFM, and MOOC, to assess the model's performance in real-world setting. Our experiment focuses on the tasks of future edge prediction, whose objective is to predict the probability of an edge occurring between two nodes at a given time.

As an additional baseline with respect to those used for the long range tasks, we consider Edge-Bank (Poursafaei et al., 2022), which is memorization-based method without learning that simply stores previously observed edges from a fixed-size time-window from the immediate past, and pre-

dicts stored edges as positive. We evaluated EdgeBank with different time windows spanning from a size of 1% of the training set to infinite size, i.e., all observed edges are stored in memory.

We performed hyper-parameter tuning via grid search by optimizing the area under the roc curve (AUC). Results for the best configuration are provided as average on 5 random initializations of the training. Appendix E provides additional experimental details. To give models a fair setting for comparison, the grid is computed considering a budget of $\sim$140k trainable parameters per model and the neighbor sampler size is set to 5.

**Results.** Table 2 reports the average test AUC on the C-TDG benchmarks. CTAN shows remarkable performance compared to literature models, ranking mainly first across datasets. CTAN achieves a score that on average is 4.7% better than its competitors. Considering only baselines with learnable parameters, our method improves the performance of the runner-up model by 1.6% of AUC. Secondly, we notice that CTAN is faster than literature models (see Appendix F). Our method has a speedup on average of $1.3\times$ to $2.2\times$ on the four benchmarks when one layer of graph convolutions is considered, and $1.5\times$ to $1.9\times$ when five layers are used.

Finally, it's worth noting that EdgeBank exhibits improved performance as the size of the time window increases. For instance, in the Wikipedia dataset, we observe that expanding the window size allows EdgeBank to obtain a significant gain of more than 20 points in AUC. It is reasonable to assume that this finding shows the importance of a non-dissipative behavior of the method even on real-world tasks, since more information need

Table 2: Mean test set AUC and std in percent averaged over 5 random weight initializations. Each model have a maximum budget of learnable weights equal to $\sim$140k. The higher, the better.

|  | Wikipedia | Reddit | LastFM | MOOC |
|---|---|---|---|---|
| EdgeBank$_{1\% \, tr \, set}$ | 71.03 | 71.92 | 77.59 | 61.29 |
| EdgeBank$_{50\% \, tr \, set}$ | 90.29 | 94.82 | 94.06 | 69.63 |
| EdgeBank$_{\infty}$ | 91.82 | 96.42 | **94.72** | 70.85 |
| DyRep | $88.64_{\pm 0.15}$ | $97.51_{\pm 0.10}$ | $77.89_{\pm 1.39}$ | $81.87_{\pm 2.47}$ |
| JODIE | $94.68_{\pm 1.05}$ | $96.34_{\pm 0.83}$ | $69.76_{\pm 2.74}$ | $81.90_{\pm 9.03}$ |
| TGAT | $94.91_{\pm 0.25}$ | $98.18_{\pm 0.05}$ | $81.53_{\pm 0.34}$ | $87.61_{\pm 0.15}$ |
| TGN | $95.60_{\pm 0.18}$ | $98.23_{\pm 0.10}$ | $79.18_{\pm 0.79}$ | $90.74_{\pm 0.99}$ |
| Our | **$97.55_{\pm 0.09}$** | **$98.61_{\pm 0.04}$** | **$83.81_{\pm 0.92}$** | **$92.47_{\pm 0.78}$** |

to be retained and propagated from the past to improve the final performance. Our results demonstrate that CTAN is able to better capture and exploit such information.

## 6 CONCLUSION

We have presented Continuous-Time Graph Anti-Symmetric Network (CTAN), a new framework based on stable and non-dissipative ODEs for learning long-range interactions in C-TDGs. Differently from previous approaches, CTAN's formulation allows scaling the radius of effective propagation of information in C-TDGs (i.e., allowing for a scalable long-range propagation in C-TDGs) and reimagines state-of-the-art static DGNs as a discretization of non-dissipative ODEs for C-TDGs. To the best of our knowledge, CTAN is the first framework to address the long-range propagation problem in C-TDGs, while bridging the gap between ODEs and C-TDGs.

Our experimental investigation reveals, at first, that when it comes to capturing long-range dependencies in a task, our framework significantly surpasses state-of-the-art DGNs for C-TDGs. Our experiments indicate that CTAN is capable of propagating relevant information incrementally across time to achieve accurate predictions, even when the model is only allowed to extract information from very local neighborhoods, i.e., by using only a single or few layers. Thus, CTAN enables scaling the extent of information propagation in C-TDG data structures without increasing the number of layers nor incurring in dissipative behaviors. Moreover, our results indicate that CTAN is quite effective across various graph benchmarks in both real and synthetic scenarios. In essence, CTAN showcased its ability to explore long-range dependencies (even with limited resources), suggesting its potential as a crucial step toward mitigating the over-squashing problem in C-TDGs.

We believe that CTAN lays down the basis for further investigations of the problem of over-squashing and long-range interaction learning in the C-TDG domain. Looking ahead to future developments, we plan to extend this study to explore alternative architectures resulting from different discretization methods, such as adaptive multi-step schemes (Ascher & Petzold, 1998). Additionally, we aim to assess the framework's impact in the realm of efficient neural networks, such as in Reservoir Computing (Nakajima & Fischer, 2021).

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

## A  PROOF OF PROPOSITION 1

Let us consider an aggregation function $\Phi$ that does not depend on the term $\mathbf{h}_u(t)$. The Jacobian matrix of Eq. 3 is defined as

$$\mathbf{J}(t) = \text{diag}\left[\sigma'\left(\mathbf{W}_t\mathbf{h}_u(t) + \Phi\left(\{\mathbf{h}_v(t), \mathbf{e}_{uv}, t_v^-, t\}_{v\in\mathcal{N}_u^t}\right) + \mathbf{b}_t\right)\right]\mathbf{W}_t. \quad (6)$$

Thus, it is the result of a matrix multiplication between invertible diagonal matrix and a weight matrix. Imposing $\mathbf{A} = \text{diag}\left[\sigma'\left(\mathbf{W}_t\mathbf{h}_u(t) + \Phi\left(\{\mathbf{h}_v(t), \mathbf{e}_{uv}, t_v^-, t\}_{v\in\mathcal{N}_u^t}\right) + \mathbf{b}_t\right)\right]$, then the Jacobian can be rewritten as $\mathbf{J}(t) = \mathbf{A}\mathbf{W}_t$.

Let us now consider an eigenpair of $\mathbf{A}\mathbf{W}_t$, where the eigenvector is denoted by $\mathbf{v}$ and the eigenvalue by $\lambda$. Then:

$$\mathbf{A}\mathbf{W}_t\mathbf{v} = \lambda\mathbf{v},$$
$$\mathbf{W}_t\mathbf{v} = \lambda\mathbf{A}^{-1}\mathbf{v},$$
$$\mathbf{v}^*\mathbf{W}_t\mathbf{v} = \lambda(\mathbf{v}^*\mathbf{A}^{-1}\mathbf{v}) \quad (7)$$

where $*$ represents the conjugate transpose. On the right-hand side of Eq. 7, we can notice that the $(\mathbf{v}^*\mathbf{A}^{-1}\mathbf{v})$ term is a real number. If the weight matrix $\mathbf{W}_t$ is anti-symmetric (i.e., skew-symmetric), then it is true that $\mathbf{W}_t^* = \mathbf{W}_t^\top = -\mathbf{W}_t$. Therefore, $(\mathbf{v}^*\mathbf{W}_t\mathbf{v})^* = \mathbf{v}^*\mathbf{W}_t^*\mathbf{v} = -\mathbf{v}^*\mathbf{W}_t\mathbf{v}$. Hence, the $\mathbf{v}^*\mathbf{W}_t\mathbf{v}$ term on the left-hand side of Eq. 7 is an imaginary number. Thereby, $\lambda$ needs to be purely imaginary, and, as a result, all eigenvalues of $\mathbf{J}(t)$ are purely imaginary.

## B  NON-DISSIPATIVENESS OVER TIME

**Definition 2** (Non-dissipativeness over time). *Let $u \in \mathcal{V}(t)$ be a node in the C-TDG at time $t$ and $o_t$ an event that occurs at node $u$ at time $t$. A DGN for C-TDGs is non-dissipative over time if, regardless of how many more events subsequently occur at $u$, the information of event $o_t$ will persist in $u$'s embedding.*

In essence, Definition 2 captures the idea that the embedding computed by a DGN for a node in a C-TDG retains the information from a specific event indefinitely, ensuring that the historical context is preserved and not forgotten despite the occurrence of additional events at that node.

To show that CTAN is non-dissipative over time, we analyze the entire system defined in Eq. 3 from a temporal perspective. Thus, Eq. 3 can be reformulated as:

$$\frac{\partial \mathbf{h}_u(t)}{\partial t} = \sigma \left( \mathbf{W}_t \psi(\mathbf{h}_u(t), \mathbf{x}_u(t)) + \Phi \left( \{ \psi(\mathbf{h}_v(t), \mathbf{x}_v(t)), \mathbf{e}_{uv}, t_v^-, t \}_{v \in \mathcal{N}_u^t} \right) + \mathbf{b}_t \right) \tag{8}$$

where $\mathbf{W}_t$ is the anti-symmetric weight matrix (as before), and $\psi$ is the function that computes the initial condition for the propagation of each event, considering the node representations computed in the previous event propagation $\mathbf{h}_u(t)$ and the current node input state $\mathbf{x}_u(t)$.

In this context, we can view the system as having $eL$ steps, where $e$ denotes the number of events and $L$ represents the number of propagation steps, i.e., the number of layers in the neural architecture. Furthermore, the input state of the node $\mathbf{x}_u(t)$ is only active during the occurrence of a new event, meaning that during the propagation of events, $\mathbf{x}_u(t)$ is set to 0. Therefore, its impact on information propagation is confined to the event's specific occurrence and does not affect each step of the propagation process.

The next proposition ensures that when the eigenvalues of the Jacobian matrix of Eq. 8 are placed only on the imaginary axes, then the ODE in Eq. 8 is non-dissipative in both space and time. Thus, guaranteeing the preservation of historical context over time and the propagation of event information through the C-TDGs.

**Proposition 2.** *he ODE in Eq. 8 is stable and non-dissipative over space and time if the resulting Jacobian matrix has purely imaginary eigenvalues, i.e.,*

$$Re(\lambda_i(\mathbf{J}(t))) = 0, \forall i = 1, ..., d.$$

For the proof, we refer the reader to (Gravina et al., 2023)(Appendix B) and substitute the Jacobian computed with respect to space with a Jacobian computed with respect to time.

We note that the non-dissipative behavior of the system in Eq. 8 is contingent on the specific definition of the function $\psi$. Varying the formulation of $\psi$ can yield to diverse behaviors, significantly impacting the system's ability to either preserve or dissipate information over time.

**Proposition 3.** *Provided that the aggregation function $\Phi$ does not depend on $\mathbf{h}_u(t)$, the Jacobian matrix resulting from the ODE in Eq. 8 has purely imaginary eigenvalues, i.e., $Re(\lambda_i(\mathbf{J}(t))) = 0, \forall i = 1, ..., d$ if the function $\psi$ is implemented as one of the following functions:*

- *addition, i.e., $\psi = \mathbf{h}_u(t) + \mathbf{x}_u(t)$;*

- *concatenation, i.e., $\psi = \mathbf{h}_u(t) \| \mathbf{x}_u(t)$;*

- *composition of tanh and concatenation, i.e., $\psi = tanh(\mathbf{h}_u(t) \| \mathbf{x}_u(t))$.*

*Proof.* Let's consider $\psi = \mathbf{h}_u(t) + \mathbf{x}_u(t)$, i.e., addition. In this case Eq. 8 can be reformulated as

$$\frac{\partial \mathbf{h}_u(t)}{\partial t} = \sigma \left( \mathbf{W}_t \mathbf{h}_u(t) + \mathbf{W}_t \mathbf{x}_u(t) + \Phi \left( \{ (\mathbf{h}_u(t) + \mathbf{x}_u(t)), \mathbf{e}_{uv}, t_v^-, t \}_{v \in \mathcal{N}_u^t} \right) + \mathbf{b}_t \right). \tag{9}$$

Considering that the Jacobian is computed with respect to $\mathbf{h}_u(t)$, and following the proof provided in Appendix A, we conclude that the Jacobian has purely imaginary eigenvalues.

Let's now consider $\psi = \mathbf{h}_u(t) \| \mathbf{x}_u(t)$, i.e., concatenation. In this case, the product $\mathbf{W}_t(\mathbf{h}_u(t) \| \mathbf{x}_u(t))$ can be decomposed as $\mathbf{K}_t \mathbf{h}_u(t) + \mathbf{V}_t \mathbf{x}_u(t)$, with $\mathbf{K}_t$ and $\mathbf{V}_t$ weight matrices. Similarly to the addition case, the Jacobian has purely imaginary eigenvalues.

Lastly, we consider the case of $\psi = tanh(\mathbf{h}_u(t) \| \mathbf{x}_u(t))$, i.e., the composition of *tanh* and concatenation. Here, Eq. 8 is

$$\frac{\partial \mathbf{h}_u(t)}{\partial t} = \sigma \Bigg( \mathbf{W}_t tanh(\mathbf{h}_u(t)) + \mathbf{V}_t tanh(\mathbf{x}_u(t)) + $$
$$+ \Phi \left( \{ tanh(\mathbf{h}_u(t) \| \mathbf{x}_u(t)), \mathbf{e}_{uv}, t_v^-, t \}_{v \in \mathcal{N}_u^t} \right) + \mathbf{b}_t \Bigg) \tag{10}$$

The Jacobian matrix is the results of the multiplication of three matrices, i.e., $\mathbf{J}(t) = \mathbf{ABW}_t$, with $\mathbf{A} = \text{diag}\left[\sigma'\left(\mathbf{W}_t tanh(\mathbf{h}_u(t)) + \mathbf{V}_t tanh(\mathbf{x}_u(t)) + \Phi(...) + \mathbf{b}\right)\right]$ and $\mathbf{B} = \text{diag}[1 - tanh^2(\mathbf{h}_u(t))]$. Thanks to the associative property of multiplication $\mathbf{J}(t) = \mathbf{ABW}_t = (\mathbf{AB})\mathbf{W}_t = \mathbf{DW}_t$, where $\mathbf{D}$ is the result of the multiplication of two diagonal matrices, thus $\mathbf{D}$ is diagonal. As detailed in the proof of Proposition 1 (Appendix A), we can conclude that the Jacobian matrix has purely imaginary eigenvalues. $\square$

As a counterexample, if $\psi = \mathbf{x}_u(t)$, Eq. 8 can result in a dissipative behavior, leading to the loss of information over time and compromising the model's ability to preserve historical context, since past node information is always discarded between new events. As a result, the function $\psi$ can function as a parameter to control the balance between the dissipative and non-dissipative behavior of CTAN.

## C  STABILITY OF THE FORWARD EULER'S METHOD

Following Ascher & Petzold (1998), the Euler's forward method applied to Eq. 3 is considered stable when $(1 + \epsilon\lambda(\mathbf{J}(t)))$ lies within the unit circle in the complex plane for all eigenvalues of the system. However, since the eigenvalues of the Jacobian matrix are exclusively imaginary, it follows that $|1 + \epsilon\lambda(\mathbf{J}(t))| > 1$, thus Eq. 3 is unstable when solved with forward Euler's method.

To enhance the stability of the numerical discretization method, we subtract a small positive constant $\gamma > 0$ from the diagonal elements of the weight matrix $\mathbf{W}$. This adjustment allows the eigenvalues of the Jacobian to possess a slightly negative real part, which positions $(1 + \epsilon\lambda(\mathbf{J}(t)))$ within the unit circle and enhancing the stability of the numerical discretization method. However, as detailed in Section 3, since $Re(\lambda_i(\mathbf{J}(t))) < 0$, the ODE becomes slightly dissipative. In conclusion, the term $\gamma$ can serve as a parameter for balancing the dissipative and non-dissipative behavior.

## D  DATASETS DESCRIPTION AND STATISTICS

Table 3 contains the statistics of the employed datasets. In the following, we describe the datasets and their generation.

Table 3: Statistics of the datasets used in our experiments. We report the total number of nodes and edges in the dataset for the temporal path graph (i.e., T-PathGraph) and temporal Pascal VOC (i.e., T-PascalVOC).

| | # Nodes | # Edges | #Edge ft. | Split | Surprise Index |
|---|---|---|---|---|---|
| T-PathGraph | 3,000-20,000 | 2,000-19,000 | 1 | 70/15/15 | 1.0 |
| T-PascalVOC$_{10}$ | 2,671,704 | 2,660,352 | 14 | 70/15/15 | 1.0 |
| T-PascalVOC$_{30}$ | 2,990,466 | 2,906,113 | 14 | 70/15/15 | 1.0 |
| Wikipedia | 9,227 | 157,474 | 172 | 70/15/15, Chronological | 0.42 |
| Reddit | 11,000 | 672,447 | 172 | 70/15/15, Chronological | 0.18 |
| LastFM | 2,000 | 1,293,103 | 2 | 70/15/15, Chronological | 0.35 |
| MOOC | 7,144 | 411,749 | 4 | 70/15/15, Chronological | 0.79 |
| tgbl-wiki-v2 | 9,227 | 157,474 | 172 | 70/15/15, Chronological | 0.108 |
| tgbl-review-v2 | 352,637 | 4,873,540 | - | 70/15/15, Chronological | 0.987 |
| tgbl-coin-v2 | 638,486 | 22,809,486 | - | 70/15/15, Chronological | 0.120 |

**Sequence classification on temporal path graphs.** To craft a temporal long-range problem, we first introduced a sequence classification problem on path graphs (Bondy & Murty, 1976), which is a simple linear graph consisting of a sequence of nodes where each node is connected to the previous one. In the temporal domain, the nodes of the path graph appear sequentially over time from first to last (e.g., bottom-to-top in Figure 4).

We define the task objective as the prediction of the feature seen in the first node (colored in orange in Figure 4) by making the prediction leveraging only the last node representation (colored in red in Figure 4) computed at the end of the sequence, i.e., when the last event appears. Note that this task is akin to the sequence classification task designed in (Chang et al., 2019), with the addition of a graph convolution. We set the feature of the first node to be either 1 or $-1$, while we set every other

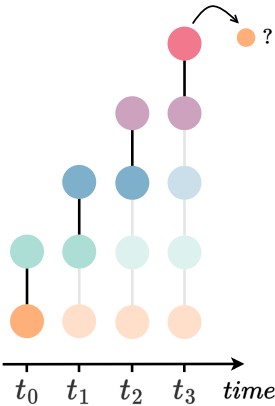

Figure 4: The illustration of the sequence classification task on a temporal path graph consisting of 5 nodes. The first node (colored in orange) has an initial feature that can be either $1$ or $-1$. All the other nodes and edges have a feature set to random value sampled uniformly in $[-1, 1]$. At the end of the sequence, the representation computed for the last node (colored in red) is used to predict the original value of the first node. At each timestamp, the faded portion of the graph corresponds to historical information.

node and edge feature to be sampled uniformly in the range $[-1, 1]$. In other words, the feature $\mathbf{x}_{u_0}$ of the first node $u_0$ contains a signal to be remembered as noise is added through the propagations steps along the graph. Formally, we create a C-TDG: $\mathcal{G} = \{o_t \,|\, t \in [t_0, t_n]\}$, such that

$$o_t = (t, \mathcal{E}_\oplus, u_t, u_{t+1}, \mathbf{x}_{u_t}, \mathbf{x}_{u_{t+1}}, \mathbf{e}_{u_t, u_{t+1}}),$$

where $\mathbf{x}_{u_0} \sim \text{Bernoulli}(0.5)^5$, and $\mathbf{x}_{u_j} \sim \mathcal{U}_{[-1,1]}, \forall j > t_0$ and $\mathbf{e}_{u_t, u_{t+1}} \sim \mathcal{U}_{[-1,1]}, \forall t$.

For this task we considered 8 temporal graph path datasets with different sizes, ranging from $n = 3$ to $n = 20$, with $n$ the number of nodes. For every graph size we generate 1,000 different graphs, and we split the dataset into train/val/test with the ratios 70%-15%-15%.

**Temporal Pascal-VOC.** We use the `PascalVOC-SP` (Dwivedi et al., 2022) dataset to design a new temporal long-range task for edge classification. `PascalVOC-SP` is a node classification dataset composed of graphs created from the images in the Pascal VOC 2011 dataset (Everingham et al., 2015). A graph is derived from each image by extracting superpixel nodes using the SLIC algorithm (Achanta et al., 2012) and constructing a `rag-boundary` graph to interconnect these nodes. Each node in a graph corresponds to one region of the image belonging to a particular class, see Figure 5 for an example. `PascalVOC-SP` contains long-range interactions between spatially distant image patches, evidenced by its average shortest path length of 10.74 and average diameter of 27.62 (Dwivedi et al., 2022).

To craft a temporal task, we consider that nodes in a `rag-boundary` graph appear from the top-left to the bottom-right of the image, sequentially. We do so by selecting the top-leftmost node, i.e., the one closest (by means of $L_1$ norm) to the origin in image coordinates. From this node, we traverse the graph with a Breadth-First-Search, visiting each node exactly once. The order of edge traversal corresponds to the timestamp of edge appearance in the temporal task. We set the task's objective to be the prediction of the class of the node that is being visited by the current edge. Note that the traversal removes a large number of edges from the initial graph, making the propagation of class information more difficult, see Figure 5.

Neighborhoods are constructed based on coordinates, connecting a node with its 8 spatially closest neighbors. Nodes have 12 features extracted by channel-wise statistics on the image (mean, std, max, min) and 2 features defining the spatial location of the superpixel; we normalize these spatial features in the [0, 1] range. We consider two SLIC superpixels compactness of 10 and 30 (smaller compactness means fewer patches). To allow for batching, we fix the number of nodes in each graph, allowing batching of edges that occur at the same timestep across different graphs together. To do so,

---

[5]Note that we sample 1 or -1 rather than 0 or 1 to make the problem balanced around zero.

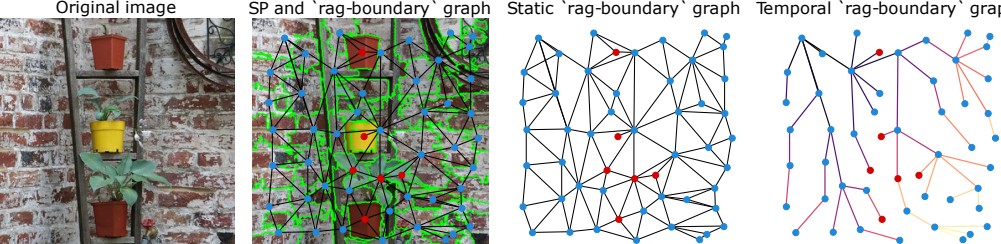

Figure 5: Construction of the Temporal `PascalVOC-SP` dataset. The SLIC algorithm extracts patches from an image. We create the `rag-boundary` graph connecting neighboring patches based on spatial closeness. We construct a temporal graph by traversing from the topleftmost node with BFS. The goal of the task is to predict the class of the destination node at each visited edge - in the figure, either 'potted plant' (red) or 'background' (blue). For clarity in this visualization, the compactness of the SLIC algorithm is low.

we discard `rag-boundary` graphs with fewer nodes than the limit, and discard excess nodes on graphs with more nodes than the limit, according to time (i.e., the most recent nodes are dropped). This removes a small number of nodes corresponding to image patches on the bottom-right of the image. In practice, for the two compactness levels 10 and 30, we set the number of minimum nodes per graph to be 434 and 474, which gives us 6,156 and 6,309 temporal graphs (out of the total 11,355 images in the dataset). The resulting temporal datasets have 2,660,352 and 2,906,113 edges respectively.

**C-TDG benchmarks.** For the C-TDG benchmarks we consider four well-known datasets proposed by Kumar et al. (2019):

- **Wikipedia**: one month of interactions (i.e., 157,474 interactions) between user and Wikipedia pages. Specifically, it corresponds to the edits made by 8,227 users on the 1,000 most edited Wikipedia pages;

- **Reddit**: one month of posts (i.e., interactions) made by 10,000 most active users on 1,000 most active subreddits, resulting in a total of 672,447 interactions;

- **LastFM**: one month of who-listens-to-which song information. The dataset consists of 1000 users and the 1000 most listened songs, resulting in 1,293,103 interactions.

- **MOOC**: it consists of actions done by students on a MOOC online course. The dataset contains 7,047 students (i.e., users) and 98 items (e.g., videos and answers), resulting in 411,749 interactions.

Since the datasets do not contain negative instances, we perform negative sampling by randomly sampling non-occurring links in the graph, as follows: (i) during training we sample negative destinations only from nodes that appear in the training set, (ii) during validation we sample them from nodes that appear in training set or validation set and (iii) during testing we sample them from the entire node set.

For all the datasets, we considered the same chronological split into train/val/test with the ratios 70%-15%-15% as proposed by Xu et al. (2020).

**Transductive vs Inductive Settings.** In Section 5.2 we employed transductive setting and random negative sampling as in Kumar et al. (2019); Xu et al. (2020); Rossi et al. (2020); Yu et al. (2023); Cong et al. (2023). We chose not to employ an inductive setting as it is not easily applicable to C-TDGs. Specifically, there is no clear consensus in the literature regarding the definition of inductive settings, making it difficult to identify the nodes considered for assessing this experimental setup (e.g. Xu et al. (2020) differs from Rossi et al. (2020)). Some definitions of inductive settings lead to the number of sampled inductive nodes to be not statistically relevant for evaluation. Other interpretations of inductive settings disrupt the true dynamics of the graph, i.e., in Rossi et al. (2020), certain nodes and their associated edges are removed from the training set with the purpose of isolating an inductive set of nodes. Thanks to the analysis performed in Yu et al. (2023), we can also observe

that among all the considered datasets in our paper there is mix of inductive and transductive edges, which can be measured with the *surprise index* from Yu et al. (2023), measuring the proportion of unseen edges at test time; reported in Table 3. Hence, achieving strong performance on tasks with a high surprise index offers valuable insights into the model's capability to address the inductive setting. Comparing CTAN performance to the surprise index, it is clear that CTAN can cope reasonably well even in fully inductive tasks (where the cold start problem might be an issue), such as those in Section 5.1 where it generally ranks first among other baselines.

## E    EXPLORED HYPER-PARAMETER SPACE

In Table 4 we report the grids of hyper-parameters employed in our experiments by each method. We recall that the hyper-parameters $\epsilon$, $\gamma$, and $\psi$ refer only to our method.

Table 4: The grid of hyper-parameters employed during model selection for the following three tasks: Sequence classification on temporal path graphs, Temporal Pascal-VOC, and Link Prediction – here, abbreviated as and color-coded in (*Seq*, orange), (*Pasc*, green), and (*Link*, blue), respectively. For *Sec* and *Pasc*, we conducted 5 runs and 10 runs with different random seeds for different weight initializations *for each configuration*, whereas for *Link*, we conducted 5 runs only for the configuration that resulted in the best performance in the initial run. For the three tasks, the models were configured to have a maximum number of learnable parameters of $\sim$20k, $\sim$40k, and $\sim$140k, respectively. Training was conducted for 20 epochs, 200 epochs, and 1000 epochs, respectively. For *Seq* and *Pasc*, we employed a scheduler *halving the learning rate* with a patience of 5 epochs, 20 epochs, respectively, whereas for *Link* we used early stopping with a patience of 50 epochs. For all tasks, the neighbor sampler size was set to 5. The batch size was set to 128, 256, and 256, respectively. We used the loss, F1-score, and AUC on the validation set to optimize for the hyper-parameters.

| Hyper-parameter | Method | Values | | |
|---|---|---|---|---|
| | | Seq | Pasc | Link |
| optimizer | | | Adam | |
| learning rate | | $3 \cdot 10^{-4}$ | $3 \cdot 10^{-4}$ | $10^{-4}, 10^{-5}$ |
| weight decay | | $10^{-7}$ | $10^{-5}$ | $10^{-6}$ |
| n. GCLs | | | 1, 3, 5 | |
| $\sigma$ | | | tanh | |
| $\epsilon$ | | $1, 0.5, 10^{-1}, 10^{-2}$ | $1, 0.5, 10^{-1}, 10^{-2}$ | $0.5, 10^{-1}, 10^{-2}, 10^{-3}$ |
| $\gamma$ | | $1, 0.5, 10^{-1}, 10^{-2}$ | $1, 0.5, 10^{-1}, 10^{-2}$ | $0.5, 10^{-1}, 10^{-2}, 10^{-3}$ |
| $\psi$ | | | concat, $\psi = \tanh(\mathbf{h}^{i-1}(t_e)||\mathbf{x}(i))$ | |
| time dim | | 1 | 1 | 16 |
| | TGN | 19, 9 | 21, 10 | 33, 20 |
| | DyRep | 53, 26 | 74, 37 | 118, 87 |
| memory dim (= DGN dim) | JODIE | 69, 34 | 97, 48 | 164, 122 |
| | TGAT | 24, 12 | 24, 12 | 33, 23 |
| | CTAN | 53, 26 | 74, 37 | 128, 96 |

## F    COMPLETE RESULTS

In Table 5 we report the results on the sequence classification task on temporal path graphs, and in Table 6 we show the complete results on the link prediction task, including the performance of Edge-Bank with different time window sizes. In this scenario, we observe that EdgeBank is particularly good at capturing long-range information along the time dimension in the LastFM task, surpassing all the baselines and CTAN as the time window increases. We highlight that the experiments in Section 5.2 are meant to outline how CTAN outperforms baselines under an *even field* of number of trainable parameters (i.e., 140k) and restricted range of hyper-parameter values, e.g., sampler size equal to 5. On the other hand, EdgeBank is a non-parametric method that at the time of inference accesses the entire temporal adjacency matrix. In LastFM, the median node degree after training is 903 (mean $1152 \pm 1722$), which is relatively high compared to other datasets. For the average node in LastFM, EdgeBank pools information from 903 node neighbors, while the setting in Section 5.2 allows baselines to pool information from 5 randomly sampled neighbors. As nodes have larger

Table 5: Results of the sequence classification on path graph long-range task, for increasing graph length $n$. The performance metric is the mean test set accuracy score, averaged over 10 different random weights initializations for each model configuration. Models have a maximum budget of learnable parameters equal to $\sim$20k.

| | $n=3$ | $n=5$ | $n=7$ | $n=9$ | $n=11$ | $n=13$ | $n=15$ | $n=20$ |
|---|---|---|---|---|---|---|---|---|
| DyRep | $100.0_{\pm 0.0}$ | $49.2_{\pm 2.1}$ | $51.0_{\pm 1.76}$ | $47.93_{\pm 2.73}$ | $44.87_{\pm 0.89}$ | $46.73_{\pm 1.55}$ | $48.6_{\pm 2.48}$ | $50.47_{\pm 2.88}$ |
| JODIE | $100.0_{\pm 0.0}$ | $100.0_{\pm 0.0}$ | $100.0_{\pm 0.0}$ | $100.0_{\pm 0.0}$ | $98.53_{\pm 4.64}$ | $97.4_{\pm 7.99}$ | $60.0_{\pm 14.91}$ | $50.87_{\pm 2.46}$ |
| TGAT | $100.0_{\pm 0.0}$ | $100.0_{\pm 0.0}$ | $50.67_{\pm 4.12}$ | $47.87_{\pm 2.72}$ | $42.67_{\pm 2.15}$ | $43.53_{\pm 0.83}$ | $50.53_{\pm 2.15}$ | $49.07_{\pm 1.55}$ |
| TGN | $100.0_{\pm 0.0}$ | $100.0_{\pm 0.0}$ | $60.2_{\pm 13.2}$ | $48.13_{\pm 1.63}$ | $45.07_{\pm 1.64}$ | $44.4_{\pm 0.64}$ | $48.67_{\pm 2.76}$ | $50.13_{\pm 2.17}$ |
| Our | $100.0_{\pm 0.0}$ | $100.0_{\pm 0.0}$ | $100.0_{\pm 0.0}$ | $99.93_{\pm 0.21}$ | $99.6_{\pm 0.56}$ | $98.67_{\pm 1.89}$ | $93.47_{\pm 8.78}$ | $88.93_{\pm 12.06}$ |

degrees, sampling larger neighborhoods is fundamental to access and therefore retain information. To show that CTAN performance is limited by the considered range of hyper-parameter values, we present in Table 6 the performance of CTAN by solely adjusting the neighbor sampler size, while maintaining a budget of $\sim$350k learnable parameters. The evaluation involves substituting various sampler size values into the optimal combination of hyper-parameters obtained for CTAN on the LastFM dataset (as in Section 5.2), with the embedding dimension configured to achieve the target of $\sim$350k learnable parameters (i.e., 192). The results indicate that CTAN performs better by adjusting the sampler size alone.

We report the average time per epoch (measured in seconds) for each model on the four considered link prediction datasets in Table 7. In this evaluation, each model has the same embedding dimension and number of GCLs. Similarly, Figure 6 shows the average time per epoch of each model on the Wikipedia dataset. Here, the time is reported with respect to a varying embedding size and similar number of GCLs.

Table 6: Mean test set AUC and std in percent averaged over 5 random weight initializations. Each model have a maximum budget of learnable weights equal to $\sim$140k. The higher, the better.

| | Wikipedia | Reddit | LastFM | MOOC |
|---|---|---|---|---|
| EdgeBank$_{1\% \, tr \, set}$ | 71.03 | 71.92 | 77.59 | 61.29 |
| EdgeBank$_{5\% \, tr \, set}$ | 81.65 | 85.07 | 86.75 | 63.93 |
| EdgeBank$_{10\% \, tr \, set}$ | 85.26 | 89.07 | 89.87 | 65.18 |
| EdgeBank$_{25\% \, tr \, set}$ | 88.31 | 92.92 | 92.74 | 67.49 |
| EdgeBank$_{50\% \, tr \, set}$ | 90.29 | 94.82 | 94.06 | 69.63 |
| EdgeBank$_{75\% \, tr \, set}$ | 91.11 | 95.63 | 94.55 | 70.46 |
| EdgeBank$_{100\% \, tr \, set}$ | 91.52 | 96.08 | 94.69 | 70.80 |
| EdgeBank$_{\infty}$ | 91.82 | 96.42 | 94.72 | 70.85 |
| DyRep | $88.64_{\pm 0.15}$ | $97.51_{\pm 0.10}$ | $77.89_{\pm 1.39}$ | $81.87_{\pm 2.47}$ |
| JODIE | $94.68_{\pm 1.05}$ | $96.34_{\pm 0.83}$ | $69.76_{\pm 2.74}$ | $81.90_{\pm 9.03}$ |
| TGAT | $94.91_{\pm 0.25}$ | $98.18_{\pm 0.05}$ | $81.53_{\pm 0.34}$ | $87.61_{\pm 0.15}$ |
| TGN | $95.60_{\pm 0.18}$ | $98.23_{\pm 0.10}$ | $79.18_{\pm 0.79}$ | $90.74_{\pm 0.99}$ |
| Our | $97.55_{\pm 0.09}$ | $98.61_{\pm 0.04}$ | $83.81_{\pm 0.92}$ | $92.47_{\pm 0.78}$ |

# G RESULTS ON TGB BENCHMARKS

We evaluate CTAN on the Temporal Graph Benchmark (TGB) (Huang et al., 2023). TGB contains a set of real-world small-to-large scale benchmark datasets with varying graph properties. While TGB contains two different graph tasks, namely dynamic link property prediction and dynamic node property prediction, for consistency with the rest of the presentation in this paper we focus on the former. To overcome the existing limitations on negative edge sampling, i.e., where only one random negative edge is sampled per each positive edge, TGB provides pre-sampled negative edge sets with both *random* and *historical* negatives (Poursafaei et al., 2022). Here, for each positive

Table 7: Mean time (in seconds) and std averaged over 10 epochs. Each model is run with an embedding dimension equal to 100 on an Intel(R) Xeon(R) Gold 6278C CPU @ 2.60GHz. The subscript $5L$ means that the model has 5 GCLs, while $1L$ means 1 layer.

|  |  | Wikipedia | Reddit | LastFM | MOOC |
|---|---|---|---|---|---|
| 1 layer | DyRep | $27.07_{\pm 0.32}$ | $161.43_{\pm 0.96}$ | $216.88_{\pm 2.83}$ | $53.32_{\pm 0.56}$ |
|  | JODIE | $20.62_{\pm 0.24}$ | $131.71_{\pm 0.85}$ | $176.61_{\pm 3.02}$ | $43.92_{\pm 0.68}$ |
|  | TGAT | $11.56_{\pm 0.14}$ | $67.83_{\pm 0.64}$ | $139.79_{\pm 20.78}$ | $\mathbf{33.92}_{\pm 0.50}$ |
|  | TGN | $30.92_{\pm 0.25}$ | $196.87_{\pm 1.35}$ | $289.22_{\pm 30.38}$ | $53.46_{\pm 0.62}$ |
|  | Our | $\mathbf{11.16}_{\pm 0.11}$ | $\mathbf{64.48}_{\pm 0.56}$ | $\mathbf{123.19}_{\pm 11.33}$ | $34.42_{\pm 0.50}$ |
| 5 layer | TGAT | $101.26_{\pm 0.46}$ | $895.35_{\pm 5.46}$ | $862.47_{\pm 217.38}$ | $73.77_{\pm 1.29}$ |
|  | TGN | $127.99_{\pm 0.60}$ | $1099.19_{\pm 3.91}$ | $1034.24_{\pm 221.04}$ | $95.45_{\pm 1.07}$ |
|  | Our | $\mathbf{60.16}_{\pm 0.20}$ | $\mathbf{532.36}_{\pm 9.87}$ | $\mathbf{495.18}_{\pm 111.13}$ | $\mathbf{56.19}_{\pm 0.63}$ |

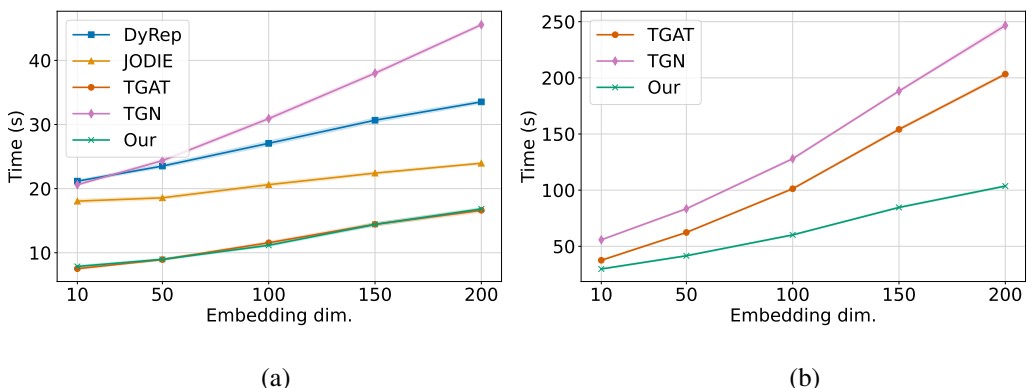

(a)          (b)

Figure 6: Average time per epoch (measured in seconds) and std with respect to the embedding size computed on the Wikipedia dataset, averaged over 10 epochs. The experiments were carried out on an Intel(R) Xeon(R) Gold 6278C CPU @ 2.60GHz. On the left (a), each model has 1 DGN layer (when possible), while on the right (b) the models have 5 GCLs.

edge, several negatives are sampled for the evaluation (Huang et al., 2023). Please refer to (Huang et al., 2023) for more information on datasets and tasks.

**Experimental Setup.** We adapt and re-use the TGB training loop and evaluation loop to fit our framework, and we select three datasets for measuring CTAN performance: *tgbl-wiki-v2, tgbl-review-v2, tgbl-coin-v2*. Because of the large dimension of many of the datasets, we can not perform model selection as extensive as that in the main paper (i.e., Section 5). In fact, the larger dataset we consider, i.e., tgbl-coin-v2, has roughly 40 times as many edges as the largest real dataset considered in Section 5 (i.e., LastFM). For these reasons, we limit to a reasonable configuration of CTAN, reported in Table 9. The experiments are executed on two NVIDIA Tesla V100 GPUs. Under such configuration and hardware, the wall time of one training epoch is approximately 20s, 11m, 1h2m, and the time of one validation pass (because of the large negative edge sets) is 1m32s, 1h12m, 5h20m, respectively for tgbl-wiki-v2, tgbl-review-v2, tgbl-coin-v2.

**Results.** To avoid having an overly unbalanced setup in terms of the number of trainable parameters, in Table 10, we report the ratio between the test Mean Reciprocal Rank (MRR) score in percentage points and the employed total number of trainable hidden units, i.e., MRR × 100/Ntot, for a more fair evaluation. As it can be seen, CTAN generally outperforms the baselines. Moreover, even considering the un-normalized (i.e., original) MRR, we observe that, even with fewer trainable parameters, CTAN achieves on par or better performance compared to various literature methods. We also note that, considering the original MRR, CTAN performs quite well on review-v2, even significantly outperforming state-of-the-art methods DyGFormer (Yu et al., 2023) and GraphMixer (Cong et al., 2023). In such dataset the *surprise index* (Poursafaei et al., 2022) is 0.987, meaning that nodes do not have large histories. In this case, it seems that CTAN better propagates information from neighbors compared to methods focusing on first-hop information passing such as Cong et al. (2023) and

Table 8: Mean test set AUC and std on LastFM (in percent) for increasing size of sampled neighbors, averaged over three different weights initializations. The model has a budget of learnable weights equal to $\sim$350k.

| sampler size | 2 | 8 | 16 | 32 | 64 | 128 |
|---|---|---|---|---|---|---|
| CTAN | $82.64_{\pm 0.93}$ | $86.21_{\pm 0.58}$ | $86.16_{\pm 0.55}$ | $86.27_{\pm 0.55}$ | $86.32_{\pm 0.81}$ | $\mathbf{87.82}_{\pm 0.42}$ |

Yu et al. (2023). On the other hand, it seems that the model in (Yu et al., 2023) is well suited in propagating long-range *time* information by modeling a large number of previous node interactions within the transformer input sequence, given enough computational budget. Nevertheless, we notice that even with limited number of parameters, CTAN can be competitive within the leaderboard. We leave CTAN experiments on (i) larger number of parameters (ii) additional benchmarks and (iii) comprehensive model selection as future work.

Table 9: The grid of hyper-parameters employed during model selection for CTAN on the Dynamic Link Property Prediction task on the three TGB benchmark datasets considered: tgbl-wiki-v2, tgbl-review-v2, tgbl-coin-v2. For tgbl-wiki-v2 we conducted 5 runs with different random seeds for different weight initializations for each configuration, whereas for tgbl-review-v2 and tgbl-coin-v2, which are one order of magnitude larger, we conducted 3 different runs. The rest of the training configuration is taken from the TGB codebase: batch size is 200, weight decay penalty was 0, the optimized metric is Mean Reciprocal Rank and is evaluated with the TGB evaluator.

| Hyper-parameter | tgbl-wiki-v2 | tgbl-review-v2 | tgbl-coin-v2 |
|---|---|---|---|
| optimizer | | Adam | |
| $\sigma$ | | tanh | |
| $\gamma$ | | 0.1 | |
| $\psi$ | | concat | |
| n. GCLs | | 1 | |
| $\epsilon$ | | 1.0 | |
| embedding dim | | 128 | |
| sampler size | | 32 | |
| learning rate | 0.0003 | $3 \cdot 10^{-6}$ | 0.0003 |
| epochs | 200 | 30 | 20 |
| LR scheduler patience | 20 | 2 | 2 |

# H    SCALABILITY OF CTAN

Here we conduct an investigation on the scalability property of CTAN. Note that while in some related works the term scalable refers to the computational complexity of methods, here we use *scalable* to refer to how the range of information propagation can be controlled by increasing the number of graph convolutions in CTAN. To show this property, we show the results of the task in Section 5.1.1 but we isolate models based on the number of GCLs they are using. We report the results in Figure 7, which shows how for increasing GCLs, CTAN is capable of conveying information further away in the graph compared to other graph convolutional based models.

Table 10: Results of the Dynamic Link Property Prediction task on the TGB benchmark datasets (Huang et al., 2023). The table reports the normalized MRR on the test split of the datasets. The subscript contains the original MRR scores averaged over different weight initializations. All baselines' results are taken from (Yu, 2023). For CTAN, the average is taken over five (or three, marked by ‡) runs with different random seeds for different weight initializations. The number of parameters is computed from the TGB Baselines repository Huang et al. (2023) by loading the best performing model across the model selection search.

| | n. params | tgbl-wiki-v2 | tgbl-review-v2 | tgbl-coin-v2 |
|---|---|---|---|---|
| EdgeBank$_\infty$ | - | (52.50) | (2.29) | (35.90) |
| EdgeBank$_{tw-ts}$ | - | (63.25) | (2.94) | (57.36) |
| EdgeBank$_{re}$ | - | (65.88) | (2.84) | (59.15) |
| EdgeBank$_{th}$ | - | (52.81) | (1.97) | (43.36) |
| CAWN | 4M | 1.83e-05$_{(73.04\pm0.60)}$ | 4.83e-06$_{(19.30\pm0.10)}$ | — |
| DyRep | 700k | 7.42e-05$_{(51.91\pm1.95)}$ | 5.72e-05$_{(40.06\pm0.59)}$ | 6.46e-05$_{(45.20\pm4.60)}$ |
| GraphMixer | 600k | 9.96e-05$_{(59.75\pm0.39)}$ | 6.15e-05$_{(36.89\pm1.50)}$ | 1.26e-04$_{(\mathbf{75.57\pm0.27})}$ |
| DyGFormer | 1.1M | 7.26e-05$_{(\mathbf{79.83\pm0.42})}$ | 2.04e-05$_{(22.39\pm1.52)}$ | 6.83e-05$_{(75.17\pm0.38)}$ |
| JODIE | 200k | 3.15e-04$_{(63.05\pm1.69)}$ | **2.07e-04**$_{(\mathbf{41.43\pm0.15})}$ | — |
| TCL | 900k | 8.68e-05$_{(78.11\pm0.20)}$ | 1.83e-05$_{(16.51\pm1.85)}$ | 7.63e-05$_{(68.66\pm0.30)}$ |
| TGAT | 1.1M | 5.45e-05$_{(59.94\pm1.63)}$ | 1.79e-05$_{(19.64\pm0.23)}$ | 5.54e-05$_{(60.92\pm0.57)}$ |
| TGN | 1M | 6.89e-05$_{(68.93\pm0.53)}$ | 3.75e-05$_{(37.48\pm0.23)}$ | 5.86e-05$_{(58.60\pm3.70)}$ |
| Our | 200k | **3.21e-04**$_{(64.29\pm0.90)}$ | 2.03e-04$_{(40.67\pm0.10)}$[‡] | **3.64e-04**$_{(72.83\pm0.52)}$[‡] |

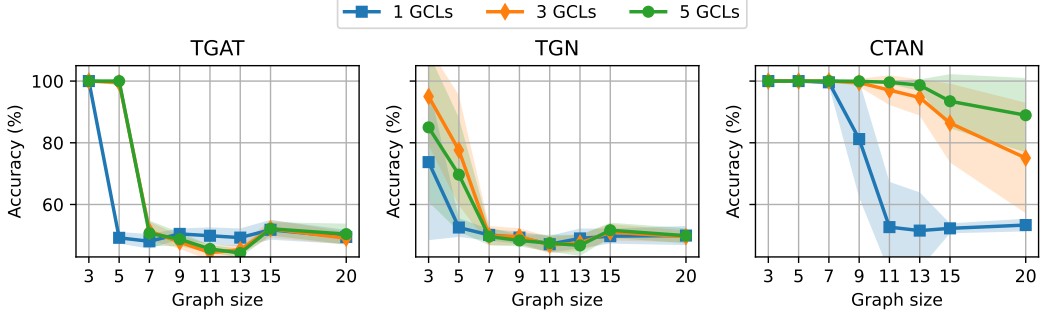

Figure 7: Mean accuracy on the T-PathGraph task on the experiment of Section 5.1.1, with distinction between the performance at different number of GCLs, for three graph convolution-based models, TGAT, TGN and CTAN. The plots show that not only CTAN can better retain information at low number of GCLs, but also that increasing the number of GCL enables solving the T-PathGraph task on longer graphs, where the task is harder because information needs to be propagated further away. The number of GCL allows CTAN to *scale* up the range of information propagation.

