# OpenReview forum: "Scalable Long Range Propagation on Continuous-Time Dynamic Graphs"
_ICLR.cc/2024/Conference — Submitted to ICLR 2024_

### Official Review · Reviewer_Bq6a · 2023-10-30

**Soundness:** 3 good
**Presentation:** 3 good
**Contribution:** 3 good
**Rating:** 6
**Confidence:** 3

**Summary:**

The paper presents Continuous-Time Graph Anti-Symmetric Network as a new graph neural network for Continuous-Time Dynamic Graphs. The authors build upon ideas presented in the literature that oversmoothing can be avoided if the eigenvalues of the dynamics are chosen carefully -- thereby facilitating long-range information transfer.

**Strengths:**

The paper is well written, and clearly articulates an important problem for GNNs as well as a potential solution.
The proposed solution is not entirely new, but appears to be well implemented.

**Weaknesses:**

- While the benefits of having a non-dissipative dynamics is discussed, the disadvantages are not discussed very prominently.
If all modes are essentially undampended there is also no filtering of noise -- the tradeoffs here should be discussed. Even better would be an experiment that considers the robustness of these ideas when faced with (adversial or even just random) noise.
- Relating to the previous point, the authors actually have added a dissipative term (-yI) when discretizing their dynamics via an Euler discretization -- I think they should discuss the importance of this term.
- Proposition 1 is essentially standard linear algebra; calling this a proposition and adding a formal proof seems to be somewhat disproportionate to what this does -- I suggest the author can simply discuss this.
- The temporal aspects of the data are almost never used; the authors simply use a "temporal" interpretation -- this raises some questions:
a) why not compare with other (non-temporal) architectures as baselines that can deal with this kind of data?
b) what benefits does the architecture really bring to temporal graphs? It appears to me that the contribution provided here is in some sense orthogonal to the fact that temporal graphs are considered as data?!
Please discuss these aspects further.

**Questions:**

See above.

In particular, I would encourage the authors to more clearly articulate the trade-offs that come with imaginary eigenvalues (e.g., while no energy is dissipated but oscillations can arise) and how they address this trade-off.
Moreover, it seems that the temporal aspects of the data is actually not really central to the exposition at hand -- in what parts does this play a strong role? This should be discussed more.

---

> ### Author Response · Authors · 2023-11-18
> **Reply to Reviewer Bq6a (part 1/3)**
>
> Our first acknowledgement goes to the reviewer for the insightful feedback. In the following, we attend to each of the raised concerns one by one.
>
> ___"While the benefits of having a non-dissipative dynamics is discussed, the disadvantages are not discussed very prominently. If all modes are essentially undampended there is also no filtering of noise -- the tradeoffs here should be discussed. Even better would be an experiment that considers the robustness of these ideas when faced with (adversial or even just random) noise."___
>
> We thank the reviewer for the comment, we have clarified this aspect in Appendix B and C.
> We agree with the reviewer that a fully non-dissipative dynamic may not be ideal for every circumstance. For such a reason, it is possible to modulate CTAN's non-dissipative behavior, either by adjusting the $\gamma$ value (as elaborated in Appendix C) or by introducing a dissipative $\psi$ function (as discussed in Appendix B).
>
> The reviewer makes a good point in suggesting testing the model under noise. Nevertheless, we believe that in our setup, where we train a model in a self-supervised way with backpropagation, the model should learn to only propagate the meaningful information while discarding the noise. To the reviewer’s comment, not all nodes are undampened, only those that contribute to good predictions during training. This in turn means the model learns to compute robust representations of nodes that only incorporate the meaningful information. While we can not investigate this in depth in the limited rebuttal time, we would point out that the experiments in Section 5.1.1 specifically test the model’s proficiency in propagating long-range information under random noise injection. In the experiment, all the intermediate information found in the linear graph consists of uniformly sampled feature values, and the model needs to be able to ignore it to correctly predict what was seen on the first node. The experiment shows that CTAN can filter out noise and learn to propagate only the meaningful information, thus demonstrating robustness to the injection of noise while propagating event information.
>
> Testing whether the model can recover meaningful node representations after seeing adversarial inputs is an interesting future direction, but we will not be able to address it in this paper.
>
> ___"Relating to the previous point, the authors actually have added a dissipative term $(-\gamma \mathbf{I})$ when discretizing their dynamics via an Euler discretization -- I think they should discuss the importance of this term."___
> We have clarified the importance of the term $(-\gamma \mathbf{I})$ in the Euler’s forward discretization in Appendix C. This term plays a crucial role in ensuring the stability of Euler’s discretization. As outlined in [1], the Euler’s forward method is considered stable when $(1+\epsilon\lambda(\mathbf{J}(t)))$ lies within the unit circle in the complex plane for all eigenvalues of the system. However, since the eigenvalues of the Jacobian matrix are exclusively imaginary, it follows that $|1+\epsilon\lambda(\mathbf{J}(t))| >1$, which makes the Eq. 3 unstable when solved using forward Euler’s method. Under this circumstance, we opted to introduce a dissipative term to facilitate the stability of the discretization method.
> We note that higher values of $\gamma$ reduce the fully non-dissipative propagation behavior of CTAN, thereby enhancing its applicability to a broader range of situations where a fully non-dissipative dynamic is not essential.
>
> [1] Computer Methods for Ordinary Differential Equations and Differential-Algebraic Equations, Ascher et al., 1998
>
> ___"Proposition 1 is essentially standard linear algebra; calling this a proposition and adding a formal proof seems to be somewhat disproportionate to what this does -- I suggest the author can simply discuss this."___
>
> We appreciate the reviewer’s suggestion regarding Proposition 1. Our primary intention was to provide a clear and thorough explanation of the proposition, and we believe that the added proof contributes to achieving this goal. Nevertheless, we are open to removing the formal proof if the reviewer believes that a simpler discussion will improve the quality of our work.

---

> > ### Author Response · Authors · 2023-11-18
> > **Reply to Reviewer Bq6a (part 2/3)**
> >
> > ___"The temporal aspects of the data are almost never used; the authors simply use a "temporal" interpretation -- this raises some questions: a) why not compare with other (non-temporal) architectures as baselines that can deal with this kind of data? b) what benefits does the architecture really bring to temporal graphs? It appears to me that the contribution provided here is in some sense orthogonal to the fact that temporal graphs are considered as data?! Please discuss these aspects further.
> > "___
> >
> > We thank the reviewer for the feedback, however we hesitate to assert that the temporal aspect is not incorporated into our approach because the main framework in Eq. 2 is explicitly defined with respect to time. To enhance the handling of irregular timestamps, we chose to decompose the Cauchy problem referenced in Eq. 2 into multiple sub-problems. It is essential to note that each sub-problem still considers the temporal aspect, as every initial condition integrates information from prior computations, facilitating the propagation of information from past events. Additionally, the temporal information of each event ($\Delta t$ in the paper) plays a crucial role in updating node embeddings, as it encodes an informative representation of the elapsed time since an event occurred on the nodes.
> >
> > Regarding the comment on the comparison with non-temporal architectures, our work fits in a line of works focusing on C-TDG specifically. While a number of non-temporal (static) approaches could be adjusted to operate in the C-TDG settings (e.g., in [2] some static approaches are adapted to the temporal domain by appending an event’s time encoding to its original features), early works have shown the large performance gains that temporal architectures bring over temporally-adapted static approaches [1,2,3].
> >
> > To the comment on contributions, by proposing the use of an ODE to model C-TDG by means of truncated non-dissipative propagation of events, CTAN effectively addresses the problem of propagation of long-range information in C-TDG, a challenge unsolved in the existing literature. Similarly to other methods, parts of CTAN could be adapted to the static domain, but its non-dissipative properties would not apply to the time dimension, which in effect does not exist in static graphs. To reinforce the importance of the temporal aspect in our framework, we have included a mathematical analysis comprising a newly added proposition that clarifies the property of non-dissipativity over both time and space of the proposed CTAN model in Appendix B.
> >
> > While we agree with the reviewer that different functions can be employed to implement Eq.2 and learn the graph dynamic, we believe that non-dissipative ODEs can offer distinct advantages for addressing the unresolved challenge of long-range propagation in the C-TDG domain. Implementing Eq.2 as a non-temporal architecture will suffer for two main aspects: (i) it does not include temporal information, which is fundamental for learning the C-TDG evolution; (ii) it may be affected from over-smoothing and/or over-squashing issues, thus reducing the model’s efficacy in long-range propagation.
> > Moreover, we note that the versatility of the function \Phi within our framework enables the utilization of the aggregation scheme more adequate for the considered task, including various non-temporal DGNs. This versatility allows for a reinterpretation of those methods as instances of our own, showcasing the broad applicability and adaptability of our proposed framework.

---

> ### Author Response · Authors · 2023-11-18
> **Reply to Reviewer Bq6a (part 3/3)**
>
> ___"Q1: I would encourage the authors to more clearly articulate the trade-offs that come with imaginary eigenvalues (e.g., while no energy is dissipated but oscillations can arise) and how they address this trade-off. Moreover, it seems that the temporal aspects of the data is actually not really central to the exposition at hand -- in what parts does this play a strong role? This should be discussed more."___
>
> We have expanded on the tradeoff between dissipative and non-dissipative behavior in Appendix B and C. As reiterated earlier, we emphasize the significance of time in our framework. Eq. 2 is explicitly defined with respect to time, and while irregular timestamps are addressed through the decomposition of Eq. 2 into multiple sub-problems, the causality of events must be respected during event propagation. Therefore, temporality is a critical aspect of our framework. Propagating multiple events in parallel would lead to catastrophic performance, as each sub-problem heavily relies on previous events. In Appendix B, we provide a mathematical analysis comprising a newly added proposition that clarifies the property of non-dissipativity over both time and space of the proposed CTAN model.
>
>
> ---
> [1] - Kumar et al., Predicting Dynamic Embedding Trajectory in Temporal Interaction Networks, 2019
>
> [2] - Xu et al., Inductive Representation Learning on Temproal Graphs, 2020
>
> [3] - Rossi et al., Temporal Graph Networks for Deep Learning on Dynamic Graphs, 2020

---

> > ### Comment · Reviewer_Bq6a · 2023-11-19
> >
> > I thank the authors for their detailed responses.

---

> > > ### Author Response · Authors · 2023-11-20
> > > **Second reply to Reviewer Bq6a**
> > >
> > > We are glad that the reviewer found our responses helpful. We would appreciate if the reviewer could provide us feedback regarding potential missing elements that may further enhance our score. Therefore, if they have any further questions or need additional information, we are more than willing to provide them.

---

### Official Review · Reviewer_udjB · 2023-11-02

**Soundness:** 3 good
**Presentation:** 3 good
**Contribution:** 2 fair
**Rating:** 5
**Confidence:** 5

**Summary:**

This paper proposes a new deep learning framework called Continuous-Time Graph Anti-Symmetric Network (CTAN) to address the problem of long-range propagation in continuous-time dynamic graphs (C-TDGs). The authors propose a theoretically motivated ODE-based framework to enable effective long-range propagation in dynamic graph learning. The paper also experimentally evaluates the proposed solutions on both synthesized and real-world networks.

**Strengths:**

S1. Modeling long-range dependencies is an important problem in dynamic graph learning.

S2. The paper provides a theoretical analysis of how the anti-symmetric weight matrices ensure stability and non-dissipativeness of the ODE for long-range propagation.

S3. The paper is generally well-written and easy to follow.

**Weaknesses:**

W1. The idea of using ordinary differential equations (ODEs) to model dynamic graphs is not novel.

W2. The authors design the sequence classification on temporal path graphs to validate the algorithm. However, I have reservations about the rationale of basing the prediction of the initial node's features solely on the last node in the sequence. Such a design indeed increases the difficulty of the task, as it requires the model to effectively propagate long-distance dependencies. However, from a practical standpoint, it may not be quite reasonable, as it completely ignores the intermediate information. Moreover, a binary classification task with only two types of features is too simple.

W3. More baselines should be included in the experiments. For example, [1-3].

[1] Y. Wang, Y.-Y. Chang, Y. Liu, J. Leskovec, and P. Li. Inductive representation learning in temporal networks via causal anonymous walks. International Conference on Learning Representations, 2021.

[2] W. Cong, S. Zhang, J. Kang, B. Yuan, H. Wu, X. Zhou, H. Tong, and M. Mahdavi. Do we really need complicated model architectures for temporal networks? arXiv preprint arXiv:2302.11636, 2023.

[3] Y. Luo and P. Li. Neighborhood-aware scalable temporal network representation learning. In Learning on Graphs Conference, pages 1–1. PMLR, 2022.

**Questions:**

See W1-W3 for details.

Minor Comment

The authors claim to present a scalable method for propagating long-range dependencies in the title and introduction, but do not follow through with this assertion in the subsequent discussion. This might give the reader the impression that the author has not fully delivered on the promise of the title in the text. I would suggest the authors elucidate the method's scalability in the methodological discussion section.

---

> ### Author Response · Authors · 2023-11-18
> **Reply to Reviewer udjB (part 1/2)**
>
> We start by thanking the reviewer for their valuable comments.
>
> ___"W1. The idea of using ordinary differential equations (ODEs) to model dynamic graphs is not novel."___
>
> We agree with the reviewer that ODE-based methods exist to model dynamic graphs, however, our CTAN framework results novel with respect to state-of-the-art methods in the C-TDG domain because of its modeling of long-range dependencies. Compared to previous work, CTAN can benefit from theoretical properties which guarantee non-dissipativeness both over the time and spatial dimension. For the former, we added a mathematical analysis comprising a newly added proposition that clarifies the property of non-dissipativity over both time and space of the proposed CTAN (see Appendix B). CTAN leverages non-dissipative ODE to propagate event information in C-TDGs by means of truncated non-dissipative propagation, which is a strategy never used in literature; it allows scalable information propagation radius; it results in a lightweight architecture that does not require the composition of multiple modules to model temporal and spatial information, thereby enabling scalable computation; and it is a general framework that allows leveraging the aggregation function that is more adequate for the specific task. Moreover, our framework focuses on long-range information propagation in C-TDGs, which is an open problem in this domain. While it is true that using ODE in dynamic graphs is not novel, we believe the C-TDG settings together with CTAN  properties and the above considerations make the method’s novelty noteworthy.
>
> ___"W2. The authors design the sequence classification on temporal path graphs to validate the algorithm. However, I have reservations about the rationale of basing the prediction of the initial node's features solely on the last node in the sequence. Such a design indeed increases the difficulty of the task, as it requires the model to effectively propagate long-distance dependencies. However, from a practical standpoint, it may not be quite reasonable, as it completely ignores the intermediate information. Moreover, a binary classification task with only two types of features is too simple."___
>
> The primary goal of the sequence classification task is to demonstrate models’ capability to propagate information without dissipating it. A compelling approach to achieve this is by transferring information seen on an initial node along a linear graph, and to be able to use that information at a later point along such a graph. The reviewer is correct in pointing out that depending on the task at hand, information in C-TDGs might not always need to propagate this way, depending on how much high-order information is present in the graph and how much of it is noise. Nevertheless, we _specifically designed_ the Sequence Prediction task (Section 5.1.1) to test whether models exhibit smoothing or dissipative behavior when presented with noisy intermediate information. Models that fail at such task are incapable of propagating information far away, and therefore can not model high-order information, which CTAN is able to do.
>
> While we acknowledge the reviewer's perspective on the simplicity of a binary classification task, our empirical findings indicate that the examined literature methods exhibit notable inadequacies in effectively addressing this task: at linear graph size=7, all convolutional-based methods (DyRep, TGAT, TGN) can not predict correctly already (accuracy<60% on a balanced dataset). This corroborates the non-dissipative information propagation claims of CTAN which instead can propagate on longer sequences  with good accuracy.
>
> ___"W3. More baselines should be included in the experiments. For example, [1-3]."___
>
> As mentioned above, in response to the reviewer's valuable suggestion, we incorporated additional baselines into our experiments, and the corresponding results are now presented in Tables 9, specifically covering methods [1] and [2] pointed out in the review. We believe that this will ease the comparison with other methods providing a more comprehensive view of our proposed method's performance in comparison to the newly introduced baselines.

---

> ### Author Response · Authors · 2023-11-18
> **Reply to Reviewer udjB (part 2/2)**
>
> ___"Q1: The authors claim to present a scalable method for propagating long-range dependencies in the title and introduction, but do not follow through with this assertion in the subsequent discussion. This might give the reader the impression that the author has not fully delivered on the promise of the title in the text. I would suggest the authors elucidate the method's scalability in the methodological discussion section."___
>
> We thank the reviewer for your insightful feedback. We appreciate the observation regarding the perceived discrepancy between the claim of presenting a scalable method for propagating long-range dependencies in the title and introduction and the subsequent discussion. We concede that the term might not have been used in the most intuitive way. In a number of related works (such as [2] and [DyGF]) the term ‘scalable’ refers to the methods __resources and computational complexity__, while our primary intent was to point out how the __range__ of information propagation can be scaled up by increasing the number of CTAN convolution layers, i.e. the non-dissipative event propagation can be scaled to reach more nodes by increasing the number of layers. To elucidate this property, we add an investigation in Appendix H showing how a larger number of Graph Convolution Layers improves the performance on the Sequence Classification task of Section 5.1.1, effectively _scaling up_ the range of propagation of the information present on the first node of the linear graph. Moreover, we emphasized the assertion of scalable non-dissipative propagation in the revised paper. We hope these enhancements address your concerns and contribute to a clearer understanding of our work.
>
> [DyGF] - Yu, et al., Towards Better Dynamic Graph Learning: New Architecture and Unified Library, NeurIPS 2023

---

> ### Author Response · Authors · 2023-11-21
> **Comment to Reviewer udjB**
>
> We have not heard from the reviewer yet. If further inquiries or clarifications are needed, please do not hesitate to ask. We hope that our comprehensive responses and inclusion of additional details and experiments have contributed positively to the manuscript. If so, we kindly ask the reviewer to consider adjusting the score accordingly.

---

### Official Review · Reviewer_eCU8 · 2023-11-04

**Soundness:** 3 good
**Presentation:** 4 excellent
**Contribution:** 3 good
**Rating:** 6
**Confidence:** 3

**Summary:**

This paper presents a deep graph network called CTAN on continuous-time dynamic graphs. The CTAN model is designed within the ordinary differential equations framework that enables efficient propagation of long-range dependencies. The paper shows that the CTAN model can robustly perform stable and non-dissipative information propagation over dynamic evolving graphs. The number of ODE discretization steps allows scaling the propagation range. The paper further presents empirical results to demonstrate the effectiveness of the CTAN model.

**Strengths:**

S1. The paper is well-motivated by the need for scalable GNN models capable of capturing long-range dependencies in continuous-time dynamic graphs.

S2. The paper provides theoretical proof of the effectiveness of the proposed model.

S3. The paper empirically validates the proposed model on many graph benchmarks. The experimental results illustrate the superiority of the proposed model.

S4. The paper is generally well-written and easy to follow.

**Weaknesses:**

W1. The benchmark datasets are all of moderate size. Validating the model's performance on larger, dynamic datasets would be beneficial.

W2. None of the C-TDG benchmarks include negative instances. The authors introduce negative sampling to the benchmark datasets by randomly sampling non-occurring links in the graph. However, the distributions of negative sampling can differ significantly from uniform distributions in real-world applications. Demonstrating the performance of the proposed model in handling negative instances across various distributions would be advantageous.

W3. Some important baselines are omitted in the experiments. For example:
- CAW: Inductive representation learning in temporal networks via causal anonymous walks. ICLR 2021
- NAT: Neighborhood-aware scalable temporal network representation learning. LoG 2022.

**Questions:**

Please refer to the Weaknesses part for details.

---

> ### Author Response · Authors · 2023-11-18
> **Reply to Reviewer eCU8**
>
> To begin, we thank the reviewer for providing feedback that allow us to improve our work.
>
> ___"W1. The benchmark datasets are all of moderate size. Validating the model's performance on larger, dynamic datasets would be beneficial."___
>
> We would point out that at the time of experiments we could not find larger C-TDG benchmarks in the literature. At the time of submission, we would have hesitated to categorize the datasets used in Section 5.2 as moderate scale: to the best of our knowledge, these tasks were among the largest ones available (as substantiated by [1]).
>
> Following up on the reviewer’s point, as of today, we are now aware of a large-scale benchmark [2] to appear at NeurIPS 23, thus we have introduced these new tasks to strengthen our work (see results in Appendix G, Table 9). We observe that given the time constraints involved in conducting all the experiments and the limited timeframe of the rebuttal phase, we regret that we are unable to execute a comprehensive and equitable model selection process, comparable to the procedures applied to other methods from the literature listed in the table (this model selection process would require weeks of GPU time, see wall time for epochs in Appendix G). Despite this limitation, it is noteworthy that our method demonstrates good performance on these new benchmark datasets.
>
> ___"W2. None of the C-TDG benchmarks include negative instances. The authors introduce negative sampling to the benchmark datasets by randomly sampling non-occurring links in the graph. However, the distributions of negative sampling can differ significantly from uniform distributions in real-world applications. Demonstrating the performance of the proposed model in handling negative instances across various distributions would be advantageous."___
>
> We agree with the reviewer's insight regarding the advantages of sampling negative instances across various distributions. To address this, we extended the evaluation of CTAN to include the TGB benchmark [1]. TGB stands out not only as a large-scale benchmark for C-TDGs, but also as it presents particularly challenging tasks in which validation and test splits include the sampling of hundreds of negative edges for each positive edge from various distributions (both random and historical negatives are used). This contributes to a more generalized and realistic evaluation, contrasting with the random negative sampling approach used in our experiments of Section 5.2. The outcomes of these evaluations on the TGB benchmark are presented in Table 9, aiming to highlight CTAN's effectiveness in handling negative instances across diverse distributions.
>
> ___"W3. Some important baselines are omitted in the experiments. For example: CAW: Inductive representation learning in temporal networks via causal anonymous walks. ICLR 2021 NAT: Neighborhood-aware scalable temporal network representation learning. LoG 2022."___
>
> As the reviewer suggested, we were able to include CAWN as part of the baselines in Table 9, where we perform evaluation on TGB. Due to the limited time in the rebuttal phase, we were not able to evaluate NAT in the paper, but we highlight how Table 9 now contains even more recent methods such as DyGFormer[3] and GraphMixer[4] (from 2023); we leave further comparisons for future work.
>
> ---
> [1] Towards Better Evaluation for Dynamic Link Prediction, Poursafaei et al., 2022
>
> [2] - Huang at al., Temporal graph benchmark for machine learning on temporal graphs, NeurIPS 2023
>
> [3] - Yu, et al., Towards Better Dynamic Graph Learning: New Architecture and Unified Library, NeurIPS 2023
>
> [4] - Cong et al., Do We Really Need Complicated Model Architectures For Temporal Networks? ICLR 2023

---

> > ### Comment · Reviewer_eCU8 · 2023-11-20
> > **Response to the authors' rebuttal**
> >
> > I appreciate the authors' detailed responses. I recommend that the authors incorporate the supplementary experiments provided in their rebuttal into any future versions of this manuscript (even in the appendix considering the page limits).

---

> > > ### Author Response · Authors · 2023-11-20
> > > **Second reply to Reviewer eCU8**
> > >
> > > We appreciate the reviewer's acknowledgment of our detailed responses. Please note that the additional experiments mentioned in our rebuttal are already included in the appendix. If there are any further inquiries or clarifications needed, please do not hesitate to ask. We hope that the inclusion of these supplementary experiments in the appendix contributes positively to the manuscript, and we kindly request to consider adjusting the score accordingly.

---

### Official Review · Reviewer_tkro · 2023-11-07

**Soundness:** 3 good
**Presentation:** 4 excellent
**Contribution:** 4 excellent
**Rating:** 5
**Confidence:** 5

**Summary:**

This work proposed a graph learning framework for continuous time dynamic graph learning, which utilize ANTI-SYMMETRIC DGN[1] to help CTDG method capture long-range information, by better model the evolution of the node memory. A special designed long range memorize task shows the effectiveness of the proposed method.

[1] https://arxiv.org/abs/2210.09789

**Strengths:**

1. This paper porosed CTAN, which is a new deep graph network for learning C-TDGs based on ODEs.
2. This paper presented novel benchmark datasets specifically designed to assess the ability of DGNs to propagate information over long spatio-temporal distances within C-TDGs

**Weaknesses:**

1.	The model is an extension from existing model A-DGN to continuous dynamic graph, though the paper claims that it is the first ODE-based architecture suitable for C-TDGs, most of the original ideas are same as the previous work, and the paper fails to contribute more on the method, which makes the novelty of the paper limited.
2.	The paper's baseline models are "too old and uncompetitive", for example, there are several sequence based CTDG methods, for example, Graphmixer[1] and DyGFormer[2], that can capture long range dependency, the paper should consider compare with more up-to-date baselines to show the effectiveness.
3.	The experiment in Table 2 can not support the claim that the model can capture long-term information well, for example, the performance of LastFM with Edgebank increases by the sequence length growth, and the proposed method fails to outperform edgebank.
4.	The test setting on benchmark dataset is different to most existing methods, the paper only contains transductive setting, and neglects the inductive setting, the negative sample strategie is also different, makes me hard to compare the performance of proposed method with existing methods.

[1] https://arxiv.org/abs/2302.11636

[2] https://arxiv.org/abs/2303.13047

**Questions:**

1.	What is the difference between the proposed method and the existing method A-DGN? Does dynamic graph learning task contain more strength of using such method?
2.	As a memory based model, how can CTAN treat with inductive setting(cold start problem)?

3. Can ODE-based encoder outperform other sequence learning encoder, when comparing with sequence-based CTDG methods like DyGFormer?
4.	Since there are three negative sample strategie mentioned in Edgebank, why does the method only compare with baseline models on random negative sample?

---

> ### Author Response · Authors · 2023-11-18
> **Reply to Reviewer tkro (part 1/4)**
>
> First, we would like to thank the reviewer for the feedback. Below, we address your comments one by one.
>
> ___"The model is an extension from existing model A-DGN to continuous dynamic graph, though the paper claims that it is the first ODE-based architecture suitable for C-TDGs, most of the original ideas are same as the previous work, and the paper fails to contribute more on the method, which makes the novelty of the paper limited."___
>
> Kindly allow us to provide a more detailed explanation of the contributions and novelty in our work, emphasizing that it goes beyond being a mere extension of A-DGN.
>
> What relates CTAN to A-DGN is that A-DGN is an ODE-based model achieving non-dissipative propagation through static graphs, i.e., in the time-unaware spatial domain. With the goal of achieving non-dissipative propagation through C-TDGs, we identified and overcame challenges whose solutions we consider novel and which categorically separate CTAN from A-DGN.
>
> First of all, we note that time-aware nodes and edges combined with possibly irregularly sampled repetitive edges between the same pair of nodes natively render A-DGN (as well as other methods designed for static graphs) inapplicable to C-TDGs. While principally one could encode time as a node/edge feature in the static cases, attempts to do so have been shown to underperform temporally-aware models [5, 6]. Less trivially, non-dissipative propagation in C-TDGs cannot be achieved through mere non-dissipative propagation through space. On the contrary, non-dissipative propagation of information through time is a property unique to DGNs designed for C-TDG, necessary for their overall non-dissipativeness. To support our claims and better highlight our contributions, we added a mathematical analysis comprising a newly added proposition that clarifies the property of non-dissipativity over both time and space of the proposed CTAN model in Appendix B of our revised manuscript.
>
> Furthermore, our CTAN framework results also novel with respect to state-of-the-art methods in the C-TDG domain since it proposes the use of an ODE to propagate event information in C-TDGs by means of truncated non-dissipative propagation, which is a strategy never used in literature, allowing scalability in the information propagation radius. Moreover, CTAN results in a lightweight architecture obviating the need for complex module compositions to model temporal and spatial information, and offers a versatile framework adaptable to diverse aggregation functions tailored for specific tasks.
>
>
> ___"The paper's baseline models are "too old and uncompetitive", for example, there are several sequence based CTDG methods, for example, Graphmixer[1] and DyGFormer[2], that can capture long range dependency, the paper should consider compare with more up-to-date baselines to show the effectiveness."___
>
> We would point out that DyGFormer has been accepted at NeurIPS 2023 (happening in Dec 23) and, unfortunately, was not available at the time of submission of our work.
>
> While both DyGFormer [1] and GraphMixer [2] may have increased capability to capture long-range dependencies, this is only applicable to __time__-only dependencies, and not spatial ones. DyGFormer models long-range time dependencies on node representations by fetching up to 4096 previous interactions (i.e., neighbors) for a node (“sequence length” in the paper [1]). Similarly, GraphMixer considers up to 2000 previous interactions ($T$ in the paper). Both methods only rely on __first-hop__ neighbors information and do not consider spatial propagation of higher-order node information, which is in fact mentioned as a limitation of DyGFormer [1, Appendix A1].
>
> Comparably, CTAN is still a graph _convolution_-based model, capable of propagating information in a non-dissipative way not only over time but also over the spatial dimension of the graph, scaling the range of propagation with the number of convolutions. This property enables propagating information to neighbors beyond the first-hop ones, which in turns allows solving tasks such as those in Section 5.1.1 and 5.1.2 in the paper. To provide further empirical evidence of CTAN’s superior ability to propagate such information, we are doing our best to add GraphMixer as a baseline to the new long-range tasks proposed in Section 5.
>
> As the reviewer suggested, for a sounder comparison with the new and more competitive methods, we added experiments on CTAN results on the Temporal Graph Benchmark [3], where (i) more recent baselines (GraphMixer and DyGFormer) (ii) larger datasets and (iii) different sampling strategies are available;  these results are in Table 9 in Appendix G. Overall, we found that CTAN is competitive with these two methods even at inferior # of parameters: we outperform GraphMixer on 2/3 datasets, and we outperform DyGFormer on 1/3 datasets. We added a brief discussion with some insights on such comparison.

---

> ### Author Response · Authors · 2023-11-18
> **Reply to Reviewer tkro (part 2/4)**
>
> ___"The experiment in Table 2 can not support the claim that the model can capture long-term information well, for example, the performance of LastFM with Edgebank increases by the sequence length growth, and the proposed method fails to outperform edgebank."___
>
> We appreciate your comment regarding any potential ambiguity in the presentation of experiments in Table 2, and we have clarified this aspect in the revised paper. In particular, we employed LastFM, MOOC, Reddit, and Wikipedia to demonstrate how well the model fares against baselines in real-world scenarios under an even playground of computational budget and sampled neighbors, without specifically focusing on long-range propagation. This comprehensive evaluation showcases the model's capabilities in diverse settings. The goal of the experiments in Section 5.2 was to show that when presented with the same amount of information and similar compute budget, CTAN can beat baselines thanks to its non-dissipative propagation properties.
>
> To the reviewer’s point, the claim that CTAN can capture long-range information is supported by the theoretical analysis we have provided throughout the paper and the experimental results achieved in particular in Section 5.1.1 and 5.1.2, which are long-range tasks by design. We can explain the fact that the performance of EdgeBank on LastFM is so exceptional with the observation that a memorization-based method such as EdgeBank is able to access the entire temporal adjacency matrix at the time of inference. In LastFM specifically, node historical neighbors are particularly dense and informative: a quick investigation shows that the median node degree before testing is 903 (1152 mean, 1722 std). This means that EdgeBank stores and uses 903 points of information per node, in comparison, CTAN and the other baselines in Section 5.2 use 5 neighbors.  As nodes tend to have larger degrees, sampling larger neighborhoods is fundamental to access and therefore retain information. This is not a limitation of CTAN but a limitation of the neighborhood sampling which is used in the majority of methods in the literature [JODIE, CTDNE, TGAT, TGN, GRAPHMIXER, TCL, CAWN].

---

> ### Author Response · Authors · 2023-11-18
> **Reply to Reviewer tkro (part 3/4)**
>
> ___"The test setting on benchmark dataset is different to most existing methods, the paper only contains transductive setting, and neglects the inductive setting, the negative sample strategie is also different, makes me hard to compare the performance of proposed method with existing methods."___
>
> In Section 5.2 we employed transductive setting and random negative sampling as it is done in JODIE [4], TGAT [5], TGN [6], EdgeBank [7] and GraphMixer [2]. We chose not to employ an inductive setting in those experiments, as it is not easily applicable to this type of graphs. Specifically:
> 1) There is no clear consensus in the literature regarding the definition of inductive settings, making it difficult to identify the nodes considered for assessing this experimental setup (e.g. [6] differs from [5]).
> 2) Some definitions of inductive settings may lead to situations where the number of sampled inductive nodes are not statistically relevant for evaluation.
> 3) Other interpretations of inductive settings disrupt the true dynamics of the graph, i.e., in [6], certain nodes (and their associated edges) are removed from the training set with the purpose of isolating an inductive set of nodes.
>
> We also observe, thanks to the analysis performed in [7], that among all the considered datasets in our paper there is mix of inductive and transductive edges, which can be measured with the  surprise index from [7], measuring the proportion of unseen edges at test time, which we report in this table:
>
> |                | T-PathGraph | T-PascalVOC10 | T-PascalVOC30 | Wikipedia | Reddit | LastFM | MOOC | tgbl-wiki-v2 | tgbl-review-v2 | tgbl-coin-v2 |
> |----------------|-------------|---------------|---------------|-----------|--------|--------|------|--------------|----------------|--------------|
> | Surprise index | 1.0         | 1.0           | 1.0           | 0.42      | 0.18   | 0.35   | 0.79 | 0.108        | 0.987          | 0.120        |
>
> Comparing the method’s performance to the surprise index in this table, one can see that our method can cope reasonably well even in fully inductive tasks, such as those in Section 5.1, i.e, T-PathGraph, T-PascalVOC10 T-PascalVOC30 where it generally ranks first among other baselines. We added these considerations to the paper in Appendix D where we present the datasets properties.
>
> As mentioned above, to ease the comparison with other methods under a unified and larger setting, we also evaluate CTAN on the larger negative sample strategies included in TGB [3], and we report the results in Table 9 (Appendix G). In these experiments/datasets, the train split uses random negative sampling as has been done in the past, but during validation and test splits, for a single positive edge, _hundreds of negative edges_ are sampled from the space of possible edges. These larger negative sets form an even more general task than separating _historical_ and _inductive_ negative sampling proposed as in [7]. To conform with the TGB evaluation, these experiments measure Mean Reciprocal Rank rather than AuC, the former being a more fine-grained metric. We hope the results in Table 9 can clarify the ability of CTAN in handling negative instances across various distributions.
>
> ___"Q1: What is the difference between the proposed method and the existing method A-DGN? Does dynamic graph learning task contain more strength of using such method?"___
>
> As mentioned earlier, A-DNG presents an architecture designed for non-dissipative information propagation within static graphs, limiting its suitability only to the spatial domain. Conversely, the C-TDG setting necessitates the capacity to capture the spatio-temporal evolution of the graph and perform non-dissipative propagation across both space and time. Given that A-DGN does not account for graph evolution and is exclusively defined for spatial information propagation, it results as inadequate for effectively learning C-TDGs. On the contrary, our framework is capable to model spatio-temporal evolution of the graph, while achieving non-dissipative propagation of information in both space and time.

---

> ### Author Response · Authors · 2023-11-18
> **Reply to Reviewer tkro (part 4/4)**
>
> ___"Q2: As a memory based model, how can CTAN treat with inductive setting(cold start problem)?"___
>
> We thank the reviewer for the comment, we have clarified the considerations between inductive vs transductive settings in the response above and in the revised paper (appendix D). In practice, our method deals with inductive nodes similarly to other approaches such as TGN, JODIE, TGAT, and EdgeBank. Node embeddings initially only capture information related to the node’s input state, which in absence of node features is a null vector. As links appear and connect an un-initialized node, its embedding will assimilate information about the evolution of the C-TDG. This propagation is backed by theoretical non-dissipative properties, which enable it to capture more informative information. In practice, the experiments of Section 5.1 confirm that even when working with completely unseen (inductive) nodes for which there is no information, CTAN can beat state-of-the-art methods
>
> ___"Q3: Can ODE-based encoder outperform other sequence learning encoder, when comparing with sequence-based CTDG methods like DyGFormer?"___
>
> We thank the reviewer for the question. Our experiments show that CTAN can outperform sequence learning encoders, such as JODIE, DyRep, and TGN, but we were not able to compare with DyGFormer [1] as the method is yet to appear in Neurips 2023, and was not available at the time of experiments. Keeping in mind the reviewer’s suggestions, to ease the comparison with the most recent methods, we were able to evaluate CTAN on tasks from TGB [3] (see Table 9 in Appendix G for the results). Even in this scenario, CTAN shows good performance with respect to other methods (even at inferior # of parameters and restricted model selection process), proving the effectiveness of ODE-based encoders, and beating DyGFormer on one benchmark. Given the limited time in this rebuttal phase and the recency of the suggested DyGFormer method, we were not able to address the reviewer’s question in depth. We discussed above why DyGFormer won’t be able to propagate information along the spatial axis, but the reviewer makes a good point noting that DyGFormer performs well in propagating long-range time information.
>
> ___"Q4: Since there are three negative sample strategie mentioned in Edgebank, why does the method only compare with baseline models on random negative sample?"___
>
> As previously mentioned, in order to consider various negative sampling strategies as pointed out by the reviewer, we assess CTAN using the extensive negative edge sets provided in TGB [3]. In these experiments, the validation and test splits entail the sampling of hundreds of negative edges for a single positive edge. These expanded negative sets present a more generalized and realistic task compared to the random negative sampling employed in our experiments in Section 5.2. We report the results on such benchmarks in Table 9 with the aim of elucidating CTAN's proficiency in handling negative instances across diverse distributions.
>
> ----
> [1] - Yu, et al., Towards Better Dynamic Graph Learning: New Architecture and Unified Library, NeurIPS 2023
>
> [2] - Cong et al., Do We Really Need Complicated Model Architectures For Temporal Networks? ICLR 2023
>
> [3] - Huang at al., Temporal graph benchmark for machine learning on temporal graphs, NeurIPS 2023
>
> [4] - Kumar et al., Predicting Dynamic Embedding Trajectory in Temporal Interaction Networks, 2019
>
> [5] - Xu et al., Inductive Representation Learning on Temproal Graphs, 2020
>
> [6] - Rossi et al., Temporal Graph Networks for Deep Learning on Dynamic Graphs, 2020
>
> [7] - Poursafaei et al., Towards Better Evaluation for Dynamic Link Prediction, 2022

---

> ### Author Response · Authors · 2023-11-21
> **Comment to Reviewer tkro**
>
> We have not heard from the reviewer yet. If further inquiries or clarifications are needed, please do not hesitate to ask. We hope that our comprehensive responses and inclusion of additional details and experiments have contributed positively to the manuscript. If so, we kindly ask the reviewer to consider adjusting the score accordingly.
>
> Lastly, we would like to highlight that a supplementary evaluation of CTAN has been added in Appendix F, demonstrating that the performance of our method on the LastFM task is constrained solely by the range of hyperparameter values considered. This finding aligns with our earlier response. Notably, a 3.5% improvement in CTAN performance was achieved by merely increasing the sampler size.

---

### Author Response · Authors · 2023-11-18
**General comment**

We thank the reviewers for their feedback and comments. We have done our best to address all points below, and we will continue to update our manuscript as we conduct complementary experiments pointed out by the reviewers. Furthermore, we thank the reviewers for acknowledging positive aspects of our submission, including the importance of modeling long-range dependencies in dynamic graphs, presenting new datasets supportive of future research on DGNs for long-range tasks, our method’s effective modelling of nodes’ evolution, and the clarity and presentation of our work.

We observe that the reviewers have requested many additional experiments. Although we will do our best to include as many experiments as possible, the time required to run all the experiments, the timespan of the rebuttal phase, and the space left in the manuscript do not allow us to include them all.

We have already incorporated some of the comments and results in the manuscript (highlighted in red in this version), and we plan to continuously update the manuscript as we collect more results until the end of the rebuttal phase.

We now address each of the reviews individually below.

---

### Meta-Review · Area_Chair_1ZDn · 2023-12-05

**Metareview:**

The paper introduces the Continuous-Time Graph Anti-Symmetric Network (CTAN), a novel deep learning framework for managing long-range propagation in continuous-time dynamic graphs (C-TDGs). Utilizing an ordinary differential equations (ODE) based approach, CTAN efficiently handles dynamic graph evolution and mitigates issues like oversmoothing, ensuring stable, non-dissipative information transfer. The framework's effectiveness is demonstrated through empirical evaluations on both synthetic and real-world networks, highlighting its capability in capturing long-range dependencies and evolving node memories in dynamic graph learning.

While the proposed CTAN model shows promising results, the reviewers have identified several weaknesses that need to be addressed:

1. The paper extends the A-DGN model to continuous dynamic graphs as the first ODE-based architecture for C-TDGs; however, its novelty is limited due to its similarity to previous work and lack of significant methodological advancements.
2. The experiment needs improvement, for example, by including new baselines in the comparison.
3. Prediction based solely on the last node is unreasonable.

Based on these weaknesses, we recommend rejecting this paper. We hope this feedback helps the authors improve their work.

**Justification For Why Not Higher Score:**

No reviewers were willing to champion this paper during the discussion phase.

**Justification For Why Not Lower Score:**

N/A

---

### Decision · Program_Chairs · 2024-01-16

Reject